# Proact-VL: A Proactive VideoLLM for Real-Time AI Companions

Weicai Yan [* 1]   Yuhong Dai [* 2]   Qi Ran [2]   Haodong Li [3]   Wang Lin [1]   Tao Jin [1 4]   Xing Xie [5]   Hao Liao [2]
Jianxun Lian [5]

🌐 **Homepage:** https://proact-vl.github.io

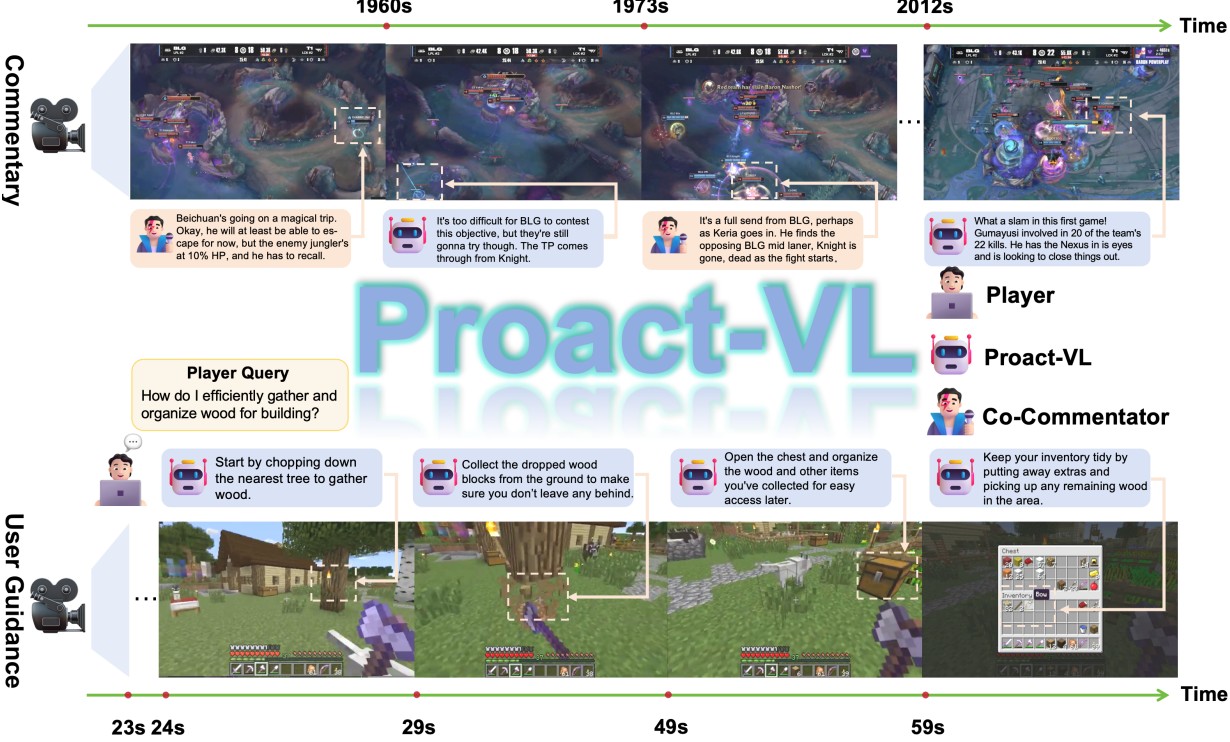

*Figure 1.* Overview of **Proact-VL**. The top section shows **Proact-VL** collaborating with other commentators for real-time commentary, while the bottom section highlights its proactive player guidance capability.

## Abstract

Proactive and real-time interactive experiences are essential for human-like AI companions, yet face three key challenges: (1) achieving low-latency inference under continuous streaming inputs, (2) autonomously deciding when to respond, and (3) controlling both quality and quantity of generated content to meet real-time constraints. In this work, we instantiate AI companions through two gaming scenarios, commentator and guide, selected for their suitability for automatic evaluation. We introduce the **Live Gaming Benchmark**, a large-scale dataset with three representative scenarios: solo commentary, co-commentary, and user guidance, and present **Proact-VL**, a general framework that shapes multimodal language models into proactive, real-time interactive agents capable of human-like environment perception and interaction. Extensive experiments show Proact-VL achieves superior response latency and quality while maintaining strong video understanding capabilities, demonstrating its practicality for real-time interactive applications.

*Equal contribution [1]Zhejiang University [2]Shenzhen University [3]South China University of Technology [4]Zhejiang University School of Software (Ningbo) Innovation and Management Center [5]Microsoft Research Asia. Correspondence to: Tao Jin <jint_zju@zju.edu.cn>, Hao Liao <haoliao@szu.edu.cn>, Jianxun Lian <jianxun.lian@microsoft.com>.

*Proceedings of the 43rd International Conference on Machine Learning*, Seoul, South Korea. PMLR 306, 2026. Copyright 2026 by the author(s).

# 1. Introduction

Recent advances in VideoLLMs (Wang et al., 2024; Bai et al., 2025a; Achiam et al., 2023; Li et al., 2025; Lin et al., 2024) have enabled AI companions that can perceive video streams and interact with users in real time for applications such as game commentary and live-stream companionship. However, effective companionship requires more than just generating appropriate responses—it demands precise control over when to speak, how long to speak, and at what pace. Constant talking can disrupt the user experience, while excessive silence undermines the sense of companionship. This highlights a key challenge for AI companions: generating controlled, short, and continuous feedback with low latency over extended interactions.

Most prior work on real-time video understanding follows a chunk-wise approach, segmenting continuous video into fixed-length chunks (typically one second) and processing them sequentially. These approaches can be broadly grouped into proactive and real-time models. Proactive models (Wang et al., 2025a; Fu et al., 2026; Liao et al., 2025; Qian et al., 2025; Yao et al., 2025; Ding et al., 2025; Wang et al., 2026; Yang et al., 2026) learn policies to decide when to respond, but typically generate complete, relatively long answers once triggered, resulting in coarse temporal granularity and higher latency. In contrast, real-time models (Chen et al., 2025; Xu et al., 2026) emphasize low-latency generation but lack explicit control over speaking behavior, often leading to excessive talking. In general, existing methods struggle to balance proactivity timing with content quality in complex real-world scenarios.

In this paper, we instantiate AI companions through two gaming applications—commentator and guide—that cover both single-assistant and multi-agent social coordination scenarios. We study three interaction settings: (1) Solo Commentary, maintaining autonomous narrative flow; (2) Co-commentary, emphasizing social coordination among multiple assistants; and (3) Real-time User Guidance, focusing on goal-directed engagement. To support training and evaluation, we construct a large-scale **Live Gaming Dataset** spanning diverse games and interaction patterns. Our framework, **Proact-VL**, as illustrated in Figure 4, introduces three key components. First, a chunk-wise input-output schema enables continuous processing of video streams. Second, a lightweight proactive mechanism autonomously decides when to respond based on visual and contextual cues. Third, a multi-tier loss function ensures stable training. The resulting system delivers low-latency, human-like interactions while maintaining video understanding capabilities. Extensive experimental results demonstrate that Proact-VL outperforms existing methods in both proactivity timing and quality. For example, on the Live Gaming Benchmark, it achieves superior scores in metrics such as TimeDiff and

F1, indicating better alignment with human commentary patterns. Additionally, Proact-VL maintains robust video understanding abilities, as evidenced by its performance on general-domain tasks. Our contributions are:

- **Dataset.** We release the **Live Gaming Dataset** (561 hours, 12 titles, three settings: solo commentary, co-commentary, user guidance), a high-quality resource for training and evaluating proactive real-time companions.

- **Method.** We propose **Proact-VL**, unifying chunk-wise processing, a proactive response mechanism, and a stability-oriented multi-tier loss to jointly control *when* and *what* to speak under streaming video constraints.

- **Results.** Proact-VL delivers strong gains in both response quality and timing while preserving general video understanding, demonstrating the practicality of always-on AI companions.

# 2. Related Work

## 2.1. Large Multimodal Models

Early multimodal LLMs (Liu et al., 2023; 2024a;b) are endowed with visual understanding by projecting visual embeddings from a pretrained vision encoder into the LLM token embedding space. This paradigm naturally extends to videos by encoding multiple frames and integrating temporal context (Lin et al., 2024; Li et al., 2025), giving rise to video large language models that support video grounded instruction following and reasoning. This line of work culminates in strong closed source systems such as GPT-4V, GPT-4o (Achiam et al., 2023), Gemini 2.5 Pro (Comanici et al., 2025), which demonstrate broad multi task multimodal understanding and instruction following capabilities. Meanwhile, open and publicly released models are rapidly advancing, including the Qwen family (Wang et al., 2024; Bai et al., 2025b;a; Xu et al., 2025a;b) and Seed1.5-VL (Team, 2025), which report competitive performance on a wide range of vision and video understanding benchmarks. Despite strong video understanding, many of these models are optimized for offline QA and still struggle with streaming video understanding.

## 2.2. Streaming and Proactive Video Understanding

Recent work has increasingly focused on processing streaming videos. A representative starting point is VideoLLM-online (Chen et al., 2024b), which reformulates training data into interleaved video chunks and text chunks, enabling a model to watch and speak in an online manner. Subsequent studies can be broadly grouped into two lines. The first line (Chen et al., 2025; Xu et al., 2026) targets streaming video understanding and low latency response generation.

LiveCC (Chen et al., 2025) scales up streaming style supervision so that the model produces sentence level outputs at a one second cadence. StreamingVLM (Xu et al., 2026) instead optimizes attention and caching to support effectively unbounded video understanding. However, these methods often provide limited control over when the model should speak. The second line (Wang et al., 2025a; Fu et al., 2026; Liao et al., 2025; Qian et al., 2025; Yao et al., 2025; Ding et al., 2025; Wang et al., 2026; Yang et al., 2026) focuses on proactive design. It typically learns a policy or a lightweight network to decide when the video stream requires a response, and once triggered, the model generates a complete answer. In practice, triggered responses tend to be lengthy and high-latency, which is unsuitable for video commentary. To address this, we design a low-latency model that decides when to respond and generates short, clip-level replies.

## 3. The Live Gaming Dataset and Benchmark

### 3.1. Video Data Collection

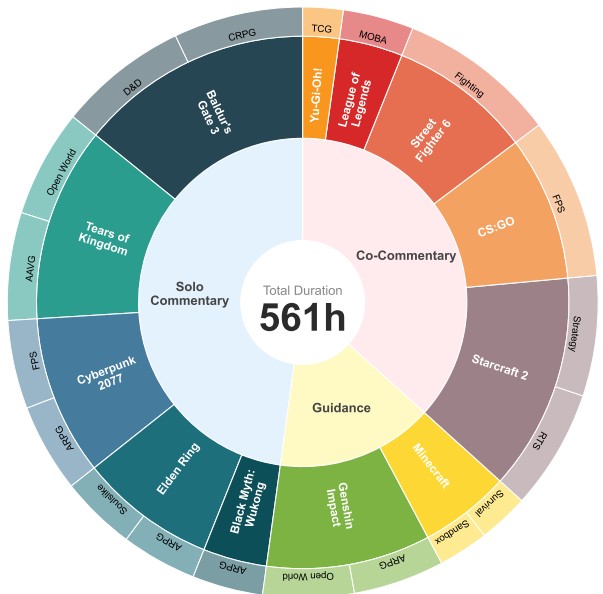

*Figure 2.* Overview of the Live Gaming Dataset. The inner, middle, and outer rings represent the three data categories, 12 specific game titles, and their corresponding genres, respectively.

Following the categorization of game genres, we selected representative popular games from each category to ensure broad coverage of the gaming landscape. We curated a diverse collection of high-traction titles spanning multiple genres and collected gameplay videos from YouTube, prioritizing content with high popularity, substantial user engagement, and high-quality commentary. To further enhance data quality, we focused on English-language videos from professional tournament broadcasts and expert influencer channels, which exhibit superior narrative density, tactical depth, and

linguistic coherence compared to casual gameplay streams. All videos were archived at a resolution of 420p, balancing visual fidelity and real-time streaming efficiency. The resulting dataset comprises 561 hours of high-quality English commentary footage across 12 blockbuster titles (see Figure 2), providing a robust multimodal foundation for proactive AI companions.

### 3.2. Data Processing

To address speaker-text alignment challenges in complex game acoustics, we develop a data processing pipeline with two branches for commentator and guide roles. Commentary videos require precise alignment of speaker identities, timestamps, and content, often disrupted by overlapping audio like background music and NPC dialogues. The pipeline diverges into specialized branches for task-specific optimization, as illustrated in Figure 3.

#### 3.2.1. COMMENTARY DATA PROCESSING

**Speech Recognition and Speaker Identification** We use WhisperX-large-v3 (Radford et al., 2023) for automated speech recognition (ASR) to extract speaker identities, timestamps, and raw transcripts. A frequency-based filter removes non-human sounds like environmental noise or NPC dialogues, isolating the primary commentary.

**Paralinguistic Nuance Labeling** To bridge the gap between static text and human-like expressivity, we leverage Qwen3-Omni-Flash to analyze audio-text pairs to label expressive elements such as pauses, laughter, and phonetic elongations, preserving natural speech prosody and affective cues.

**Domain-Specific Polishing** To address the issues of general-purpose ASR models in recognizing domain-specific terminology such as in-game mechanics and professional players, we apply a polishing stage powered by DeepSeek-V3.2-Exp (Liu et al., 2025). Conditioned on game-specific priors and predefined linguistic constraints, this stage corrects ASR-induced transcription errors, normalizes domain-specific nomenclature, and sanitizes the language data by filtering offensive or inappropriate content.

#### 3.2.2. GUIDE DATA PROCESSING

For the player guidance domain, gameplay videos are first segmented into 5-minute clips. Within each clip, Qwen3-VL-Plus is employed to identify potential player queries with their corresponding temporal intervals and to generate fine-grained, frame-aligned visual descriptions for each interval, capturing detailed player actions and scene dynamics. These descriptions are then directly provided to GPT-4.1, which filters out information irrelevant to the query and rewrites the remaining content into concise, coach-style, action-oriented guidance. This refinement step improves

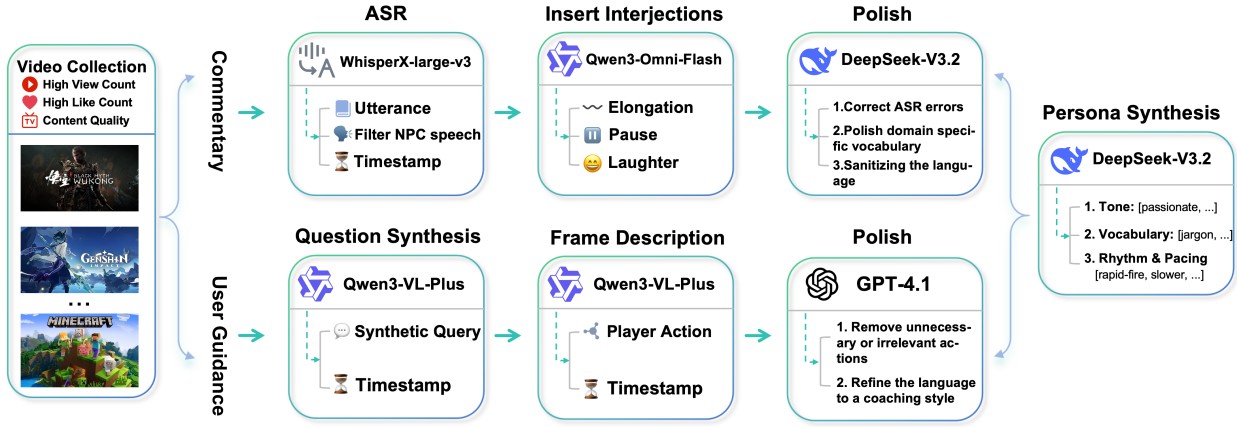

*Figure 3.* Overview of data pipeline.

clarity and professional instructional quality while preserving temporal accuracy and semantic fidelity.

### 3.2.3. PERSONA ENRICHMENT

To enhance contextual awareness and support coherent role-playing across diverse gaming scenarios, we extract structured commentator and guide personas from processed narrative data. Specifically, DeepSeek-V3.2-Exp analyzes gameplay transcripts to synthesize distinct persona profiles by identifying recurring stylistic patterns, communicative preferences, and interaction strategies, enabling consistent and context-aware commentary generation. These profiles are characterized across three dimensions: *tone* (analytical, humorous, or hype-driven), *vocabulary* (domain-specific terminology and colloquialisms), and *rhythm and pacing* (preference for exposition versus concise reactions). By encoding detailed persona information as conditioning signals during model training, we enable more consistent, contextually appropriate, and human-like interactions across diverse conversational scenarios.

### 3.3. Benchmark Construction

From our collected dataset, we build one training set and two complementary test sets: *Live Gaming Benchmark* and *Live Gaming Benchmark-Streaming*. The training set covers 10 games. *Live Gaming Benchmark* is a clip-level test suite with three subsets: (i) an in-domain subset spanning 10 games (2,640 samples) for live gaming commentary evaluation, and (ii) a common-and-general subset consisting of one common-scenario dataset (134 samples) and one out-of-domain game (240 samples). In total, it contains 3,014 clips. Live Gaming Benchmark-Streaming evaluates long-horizon stability on full-length videos by selecting one complete video per game from both Solo and Co-Commentary settings, yielding 10 videos spanning 30 minutes to 2 hours. Details are provided in the Appendix. 

## 4. Methodology

Our framework enables real-time video understanding and interaction through a novel chunk-wise processing approach combined with a proactive response mechanism. The system operates on streaming video input, making autonomous decisions about when to speak and generating appropriately timed commentary, as illustrated in Figure. 4.

### 4.1. Chunk-Wise Input Schema

To achieve real-time responsiveness, we process continuous video streams by discretizing them into fixed-duration chunks at regular intervals (one second in this paper). At each time step $t$, the model receives a chunk input triplet $(V_t, Q_t, B_t)$ that encapsulates the current context, where $V_t$ denotes the visual content occurring during the current time window, $Q_t$ denotes the optional user query providing immediate interaction context, $B_t$ denotes the environmental context including previous commentary summaries. Figure 6 provides a graphical illustration of this input structure.

The model operates causally on the streaming input, producing a chunk-wise utterance segment $U_t$ aligned with each time step $t$ in an online manner. Each segment $U_t$ is constrained to a duration suitable for real-time delivery (one second in this paper). This design enables continuous, real-time interaction where multi-segment responses flow seamlessly across consecutive chunks, allowing longer responses to naturally continue across subsequent time steps.

We implement this using a persistent transformer KV cache $\mathcal{K}_{t-1}$ that maintains keys and values from all past conditioning and generated tokens up to step $t-1$. Critically, the generated utterance $U_t$ is automatically appended to the context stream and becomes part of the input for the subsequent time step $t+1$, creating a continuous dialogue history. The complete generation process is formalized as:

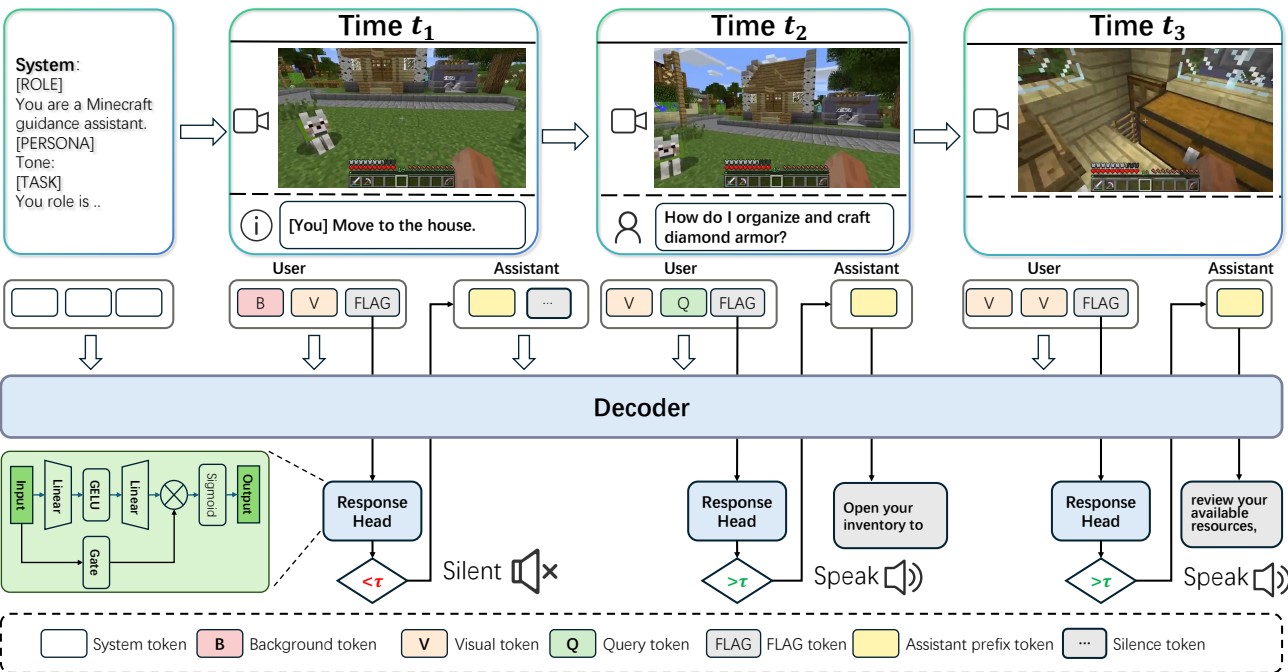

*Figure 4.* **Illustration of the Proact-VL.** At each second, Proact-VL consumes multi-source tokens (video, query, and context) and decides whether to speak by feeding the `FLAG` hidden state into a response head to obtain a score, then thresholding with $\tau$. If triggered, it appends the assistant prefix and generates a short clip-level text; otherwise, it appends the prefix with a `Silence` token to output silence.

$$(U_t, \mathcal{K}_t) = f_\theta(V_t, Q_t, B_t; \mathcal{K}_{t-1})$$

where $\theta$ denotes the model parameters and $\mathcal{K}_t$ represents the updated key-value cache for the respective token sequences. This caching mechanism enables efficient incremental processing while maintaining full temporal context, with $U_t$ serving as both the current output and future contextual input for sustained conversational coherence.

To integrate these inputs into a unified representation, we adopt a ChatML-style message format that serializes all available signals at each time step $t$ into structured role-based messages (system, user, and assistant). The complete formatting specification, including detailed tokenization rules and message sequencing, is provided in Appendix A.1.

### 4.2. Proactive Response Mechanism

Unlike conventional VLMs that respond only to explicit prompts, Proact-VL autonomously decides when to speak through a lightweight triggering mechanism. As illustrated in Figure 6, we insert a special decision token `<|FLAG|>` at the end of each user message. After processing the complete context at step $t$, we extract the hidden state $\mathbf{h}_t$ corresponding to this `<|FLAG|>` token. We then apply a minimal gated MLP head followed by a sigmoid activation to compute a speaking probability:

$$p_t = \sigma(\text{MLP}(\mathbf{h}_t)).$$

A binary decision is obtained by comparing $p_t$ against a fixed threshold $\tau$: $a_t = \mathbb{I}[p_t \geq \tau]$, where $a_t = 1$ triggers utterance generation and $a_t = 0$ maintains silence.

This lightweight gatekeeping mechanism enables real-time interaction, which is essential for maintaining natural and engaging human-AI companionship.

### 4.3. Training Strategy

We optimize the model using two complementary objectives that learn *what to say* and *when to speak*. The primary causal language modeling loss $\mathcal{L}_{\text{main}}$ supervises utterance quality, while the response loss $\mathcal{L}_{\text{resp}}$ governs speaking behavior.

Given ground-truth speaking labels $y_t \in \{0, 1\}$ derived from human commentary patterns, we apply masking to focus supervision only on positions corresponding to assistant responses. A straightforward design is to treat speaking vs. silence as a binary classification problem at each time step and optimize a per-step classification loss.

**Transition-smoothed classification loss.** We find the per-second response state should not be treated as independent points; it is a *sequence* learning problem. The dominant imbalance is not simply the number of silence vs. response seconds, but the imbalance between *state transitions* and *state persistence*. The model thus needs to learn when to *keep* the current state and when to *switch* (response↔silence).

To emphasize these rare but crucial switching steps, we use transition-aware weights $w_t$, setting $w_t = \gamma$ when $y_t \neq y_{t-1}$ and $w_t = 1$ otherwise.

The classification loss uses weighted binary cross-entropy:

$$\mathcal{L}_{\mathrm{cls}} = \frac{1}{\sum_t w_t} \sum_t w_t \left( -y_t \log p_t - (1 - y_t) \log(1 - p_t) \right).$$

**Stability regularization.** A stable and usable response mechanism further requires explicit regularization to suppress jitter and control the overall speaking rate. Thus, we introduce a regularizer that enforces both local temporal consistency and global speaking-rate constraints, local consistency promotes stability during state persistence, while the global constraint calibrates the overall response rate:

$$\mathcal{L}_{\mathrm{reg}} = \mathbb{E}\left[ (p_t - p_{t-1})^2 \mid y_t = y_{t-1} \right] + \left( \mathbb{E}[p_t] - \mathbb{E}[y_t] \right)^2,$$

where the first term encourages smooth probability transitions within continuous speaking/silence segments, and the second term regularizes the model's average speaking rate $\mathbb{E}[p_t]$ to match the human baseline $\mathbb{E}[y_t]$ (treated as constant), ensuring the AI companion speaks a similar total amount as human commentators in real-world scenarios.

The combined response loss is defined as:

$$\mathcal{L}_{\mathrm{resp}} = \mathcal{L}_{\mathrm{cls}} + \mathcal{L}_{\mathrm{reg}}.$$

The overall training objective is then formulated as:

$$\mathcal{L} = \mathcal{L}_{\mathrm{main}} + \alpha \mathcal{L}_{\mathrm{resp}}.$$

### 4.4. Infinite Inference

To support unbounded streaming under a fixed context length, we adopt a dual-cache sliding-window KV-cache mechanism. We maintain a persistent *system cache* for the initial prompt and a dynamic *streaming cache* for ongoing user/assistant tokens. When the context budget is reached, we evict the oldest portion of the streaming cache while keeping recent interactions, and apply a lightweight reverse-RoPE correction to avoid positional discontinuity. Implementation details are deferred to Appendix A.2.

## 5. Experiment

### 5.1. Setup

**Datasets.** We evaluate our model from three aspects: (1) Live Gaming Commentary, where we adopt the in-domain subset of our Live Gaming Benchmark as the primary evaluation; (2) Common and General Commentary, where we evaluate on the common-and-general subset of Live Gaming Benchmark, consisting of Ego4D Goal-Step (Song et al., 2023) as the common-scenario set and the Black Myth: Wukong as the general/out-of-domain set; and (3) Live Gaming Streaming, where we use Live Gaming Benchmark-Streaming to assess long-video streaming inference.

**Evaluation Metrics.** We evaluate our model from two complementary aspects: text quality and proactivity quality. For text quality, we report three metrics: (1) Closed Captions (CC), a win-rate metric that measures captioning quality against a closed-source model; in this paper, we compute CC as the win rate of our outputs compared with Gemini 2.5 Pro; (2) LiveU, an LLM-based metric that scores clip-level commentary quality under a streaming setting; and (3) FinalQ, which concatenates all clip outputs within a segment and evaluates the overall script quality. For proactivity timing, we use three standard metrics—TimeDiff, PAUC (Wang et al., 2025b), and F1—to measure temporal alignment and event-trigger performance. Details are provided in Appendix C.1.

**Baselines.** We consider three categories of baselines. (1) Commercial closed-source models: GPT-4o, and Gemini 2.5 Pro. (2) Proactive models: VideoLLM-online (Chen et al., 2024a), MMDuet (Wang et al., 2025a), and LiveStar (Yang et al., 2026), which first decide whether to respond and, once triggered, generate commentary. (3) Real-time models: LiveCC-7B-Base (Chen et al., 2025), LiveCC-7B-Instruct (Chen et al., 2025), and StreamingVLM (Xu et al., 2026), which emphasize low-latency streaming generation.

**Implementation Details.** We initialize our model from LiveCC-7B-Base and train it with a learning rate of 1e-5 using a cosine scheduler. We set the response-loss weight to $\alpha = 0.2$ and apply gradient clipping with `max_grad_norm = 1.0`. For the proactive response mechanism, we feed the hidden state of the `<|FLAG|>` token from the penultimate transformer layer into the response head. In our training set, the ratio between transition steps and persistence steps is approximately $1 : 5$, so we set $\gamma = 5$. We set the batch size to 64 and train for 2,000 steps, which costs approximately 200 H100 GPU-hours in total. For evaluation, we use GPT-5.1 to compute the PAUC, win-rate score(CC) and LLM-based judgment scores, including LiveU and FinalQ. Unless otherwise specified, all experiments are conducted using models fine-tuned from LiveCC-Base. For the main results reported in Secs. 5.2–5.4, we use a response threshold of $\tau = 0.3$; for all other analyses and ablations, we set $\tau = 0.5$. Additional implementation details are provided in the Appendix C.3. We also provide models initialized from Qwen-series backbones. They achieve significantly better commentary quality and proactive responses than base model, demonstrating the effectiveness of our framework (see Appendix C.4 for details).

*Table 1.* Main results of text quality on Live Gaming Benchmark. **Bold** indicates the best results, and underline the second best.

| Model | Solo Commentary | | | Co-Commentary | | | Guidance | | | Overall | | |
|---|---|---|---|---|---|---|---|---|---|---|---|---|
| | CC ↑ | LiveU ↑ | FinalQ ↑ | CC ↑ | LiveU ↑ | FinalQ ↑ | CC ↑ | LiveU ↑ | FinalQ ↑ | CC ↑ | LiveU ↑ | FinalQ ↑ |
| *Offline Models* | | | | | | | | | | | | |
| GPT-4o | 21.54 | 4.56 | 4.74 | 46.72 | 3.35 | 2.99 | **50.00** | 5.95 | **6.66** | 39.42 | 4.62 | 4.80 |
| Gemini 2.5 Pro | - | 4.87 | 5.19 | - | 3.49 | **3.59** | - | 5.73 | 5.67 | - | 4.70 | 4.82 |
| *Proactive Models* | | | | | | | | | | | | |
| VideoLLM-online | 16.71 | 4.26 | 2.07 | 17.55 | 2.57 | 1.39 | 7.08 | 3.85 | 1.75 | 13.78 | 3.56 | 1.74 |
| MMDuet | 18.62 | 2.24 | 2.32 | 24.74 | 2.59 | 2.44 | 16.88 | 3.18 | 3.28 | 20.08 | 2.67 | 2.68 |
| Livestar | 4.42 | 3.14 | 2.12 | 6.67 | 2.92 | 2.07 | 14.69 | 3.36 | 3.05 | 8.59 | 3.14 | 2.41 |
| *Real-Time Models* | | | | | | | | | | | | |
| LiveCC-7B-Base | 41.17 | 4.85 | 4.02 | 45.89 | 3.78 | 3.06 | 29.58 | 2.92 | 3.07 | 38.88 | 3.85 | 3.83 |
| LiveCC-7B-Instruct | 34.33 | 5.84 | 4.70 | 37.40 | 4.29 | 3.28 | 13.33 | 4.56 | 3.89 | 28.35 | 4.90 | 3.96 |
| StreamingVLM | 12.17 | 3.94 | 2.83 | 24.38 | 3.16 | 2.23 | 8.12 | 3.37 | 2.89 | 14.89 | 3.49 | 2.65 |
| *Real-Time Proactive Model* | | | | | | | | | | | | |
| **Proact-VL** | **53.62** | **6.89** | **5.48** | **51.46** | **5.15** | **3.59** | 42.60 | **7.52** | 6.02 | **49.23** | **6.52** | **5.03** |

*Table 2.* Main results of response quality on Live Gaming Benchmark. **Bold** indicates the best results, and underline the second best.

| Model | Solo Commentary | | | Co-Commentary | | | Guidance | | | Overall | | |
|---|---|---|---|---|---|---|---|---|---|---|---|---|
| | TimeDiff ↓ | PAUC ↑ | F1 ↑ | TimeDiff ↓ | PAUC ↑ | F1 ↑ | TimeDiff ↓ | PAUC ↑ | F1 ↑ | TimeDiff ↓ | PAUC ↑ | F1 ↑ |
| *Offline Models* | | | | | | | | | | | | |
| GPT-4o | 1.16 | **25.14** | 62.53 | 3.42 | 3.31 | 58.80 | 4.62 | **42.26** | 43.30 | 3.07 | **23.57** | 54.88 |
| Gemini 2.5 Pro | **1.03** | 22.72 | 59.02 | 2.77 | 5.38 | 56.86 | 3.96 | 25.96 | 31.82 | 2.59 | 18.02 | 49.23 |
| *Proactive Models* | | | | | | | | | | | | |
| VideoLLM-online | 10.86 | 0.04 | 7.02 | 8.95 | 0.00 | 8.16 | 17.97 | 0.00 | 4.44 | 12.59 | 0.01 | 6.54 |
| MMDuet | 27.85 | 0.00 | 0.05 | 23.54 | 0.00 | 0.16 | 28.76 | 0.41 | 0.32 | 26.72 | 0.14 | 0.18 |
| Livestar | 28.24 | 15.32 | 0.24 | 23.71 | 3.53 | 0.17 | 30.03 | 14.78 | 0.20 | 27.33 | 11.21 | 0.20 |
| *Real-Time Models* | | | | | | | | | | | | |
| LiveCC-7B-Base | 8.36 | 16.34 | 47.05 | 8.87 | 5.43 | 43.25 | 16.83 | 8.69 | 18.00 | 11.35 | 10.15 | 36.10 |
| LiveCC-7B-Instruct | 1.04 | 16.84 | 62.05 | 2.01 | 3.90 | 61.26 | 3.34 | 16.67 | 44.83 | 2.13 | 12.47 | 56.05 |
| StreamingVLM | 1.35 | 5.11 | 56.92 | 2.28 | 0.60 | 52.37 | **3.00** | 3.85 | 42.73 | 2.21 | 3.19 | 50.67 |
| *Real-Time Proactive Model* | | | | | | | | | | | | |
| **Proact-VL** | 1.20 | 20.36 | **63.25** | **0.71** | **7.01** | **77.44** | 3.21 | 26.92 | **53.91** | **1.71** | 18.10 | **64.87** |

## 5.2. Result of Live Gaming Commentary

**Text Quality.** Table 1 shows that Proact-VL achieves the best overall text quality on Live Gaming Commentary and consistently leads in Solo and Co-Commentary across CC/LiveU/FinalQ. Notably, Proact-VL is competitive with strong commercial models: it improves substantially over GPT-4o and matches/exceeds Gemini 2.5 Pro on overall LLM-judged quality, while maintaining higher CC. Its main relative weakness appears in *Guidance*: although Proact-VL remains best on LiveU, its CC/FinalQ are slightly below the strongest offline model, suggesting room to better incorporate external guidance. Overall, real-time models form the second tier, whereas prior proactive models lag significantly on both CC and LLM-judged metrics.

**Response Quality.** Table 2 further confirms Proact-VL's advantage in proactivity timing and triggering. Proact-VL achieves the best overall F1 and shows particularly strong gains in Co-Commentary and Guidance, outperforming commercial models on triggering accuracy while keeping TimeDiff low. In Solo Commentary, Proact-VL remains close to the lowest-TimeDiff commercial baselines and delivers higher F1, though its PAUC is slightly below GPT-4o, leaving headroom for sharper separation of response-worthy moments. In comparison, real-time baselines are generally stable but weaker in triggering quality, while proactive baselines suffer from severe misalignment.

## 5.3. Result of Common and General Commentary

**Text Quality.** Table 3 shows that Proact-VL consistently achieves the strongest text quality on both the *general* (Ego4D) and *unseen-game* (Black Myth Wukong) settings. On Ego4D, Proact-VL leads in overall coherence and narration quality, indicating more fluent and helpful commentary for general guidance-style streams. On Black Myth Wukong, Proact-VL remains highly competitive and attains top-tier LLM-judged text quality, suggesting strong out-of-domain

*Table 3.* Full results on text and response quality for common and general commentary. **Bold** indicates the best results, while Underline denotes the second-best results.

| Model | Ego4D | | | | | | Black Myth Wukong | | | | | |
| --- | --- | --- | --- | --- | --- | --- | --- | --- | --- | --- | --- | --- |
| | CC ↑ | LiveU ↑ | FinalQ ↑ | TimeDiff ↓ | PAUC ↑ | F1 ↑ | CC ↑ | LiveU ↑ | FinalQ ↑ | TimeDiff ↓ | PAUC ↑ | F1 ↑ |
| *Offline Models* | | | | | | | | | | | | |
| GPT-4o | 50.00 | 5.02 | 4.35 | 15.23 | 14.93 | 16.36 | 23.33 | 4.66 | 4.40 | 1.51 | **9.35** | 58.24 |
| Gemini 2.5 Pro | - | 3.32 | 3.55 | 46.21 | 15.64 | 14.55 | - | 4.97 | 5.10 | 1.06 | 8.30 | 56.00 |
| *Proactive Models* | | | | | | | | | | | | |
| VideoLLM-online | 13.06 | 3.73 | 2.28 | 16.64 | 0.80 | 17.90 | 42.50 | 4.59 | 2.92 | 11.62 | 0.00 | 6.62 |
| MMDuet | 23.88 | 3.68 | 3.28 | 35.05 | 11.24 | 7.41 | 36.46 | 2.40 | 2.31 | 28.47 | 0.00 | 0.17 |
| Livestar | 11.94 | 4.53 | 2.78 | 24.75 | 4.66 | 23.53 | 18.75 | 3.17 | 2.37 | 29.00 | 6.07 | 0.10 |
| *Real-Time Models* | | | | | | | | | | | | |
| LiveCC-7B-Base | 12.69 | 3.45 | 3.44 | 34.70 | 6.75 | 14.94 | **56.46** | 4.55 | 4.19 | 10.44 | 3.95 | 39.00 |
| LiveCC-7B-Instruct | 11.57 | 3.39 | 3.59 | 8.43 | 12.28 | 17.12 | 43.12 | 5.76 | 4.86 | **0.87** | 4.15 | 59.88 |
| StreamingVLM | 6.72 | 3.13 | 2.73 | 12.40 | 1.91 | 16.12 | 29.38 | 4.43 | 3.28 | 1.30 | 2.09 | 54.51 |
| *Real-Time Proactive Model* | | | | | | | | | | | | |
| **Proact-VL** | **63.43** | **7.21** | **5.42** | **0.32** | **33.66** | **45.82** | 55.21 | **6.22** | **5.24** | 0.90 | 8.20 | **60.06** |

generalization to a previously unseen game.

**Response Quality.** For proactive evaluation, baseline proactive models are generally unstable, especially on the unseen-game stream, often showing timing misalignment and weak triggering. In contrast, Proact-VL substantially improves proactivity timing and triggering quality in both domains: on Ego4D it achieves the best overall alignment, demonstrating reliable "when-to-speak" decisions for general guidance streams; on Black Myth Wukong it maintains strong timing and triggering performance, matching or surpassing strong real-time baselines.

*Table 4.* Text quality over time.

| Metrics | 10 | 20 | 30 | 40 | 50 | Overall |
| --- | --- | --- | --- | --- | --- | --- |
| SC | 73.75 | 75.00 | 76.25 | 78.27 | 79.48 | 82.03 |
| LiveU | 5.45 | 5.31 | 5.50 | 5.52 | 5.53 | 5.71 |
| FinalQ | 4.13 | 4.04 | 4.20 | 4.24 | 4.23 | 4.12 |

*Table 5.* Proactivity quality over time.

| Metrics | 10 | 20 | 30 | 40 | 50 | Overall |
| --- | --- | --- | --- | --- | --- | --- |
| TimeDiff | 0.70 | 0.76 | 0.92 | 0.88 | 0.82 | 0.81 |
| PAUC | 9.08 | 7.10 | 7.58 | 8.17 | 8.07 | 7.63 |
| F1 | 74.42 | 71.15 | 69.29 | 69.85 | 70.10 | 69.23 |

### 5.4. Result of Live Gaming Streaming

In this section, we evaluate the streaming inference capability of our method on Live Gaming Benchmark-Streaming. We report metrics at 10–50 minutes in 10-minute increments, as well as overall. For text quality, SC (Streaming Commentary) is defined as the predicted win rate of Proact-VL against StreamingVLM. As shown in Table 4 and 5, Proact-VL exhibits more stable long-form behavior than

StreamingVLM: its text quality remains consistent as the inference horizon increases, while response quality shows only a mild degradation and then tends to stabilize. These results demonstrate the robustness of Proact-VL for long-form streaming commentary, highlighting improved stability under sustained streaming inference.

### 5.5. Ablation Study for Training Loss

*Table 6.* Effectiveness of training loss.

| $\mathcal{L}_{cls}$ | $\mathcal{L}_{reg}$ | CC | TimeDiff | P | R | F1 |
| --- | --- | --- | --- | --- | --- | --- |
| ✓ | - | 45.54 | 18.50 | 12.13 | 14.00 | 11.03 |
| - | ✓ | 47.53 | 8.28 | 45.20 | **67.02** | 47.39 |
| ✓ | ✓ | **50.91** | **3.41** | **65.72** | 62.41 | **60.08** |

Table 6 shows that both losses contribute to training a robust response mechanism. Removing either term consistently degrades response quality: precision, recall, and F1 all drop, while proactivity timing becomes less accurate (higher TimeDiff). The impact is especially severe without $\mathcal{L}_{reg}$, where F1 decreases by 49.05 and TimeDiff increases by 15.09, and the overall CC also drops accordingly. Overall, the two losses are complementary and jointly yield the best response behavior and generation quality.

### 5.6. In-Depth Analysis

**Response Score Curve.** In Figure 5, we visualize the score curves for the first 300 seconds of a Tears of the Kingdom sample under three trigger thresholds (0.1, 0.5, and 1.0). We observe that a very low threshold 0.1 leads to near-continuous triggering and highly oscillatory scores, indicating severe over-triggering and large mismatches with labeled silence. In contrast, a high threshold 1.0 results in all-silence behavior, yet the curve remains smooth and still

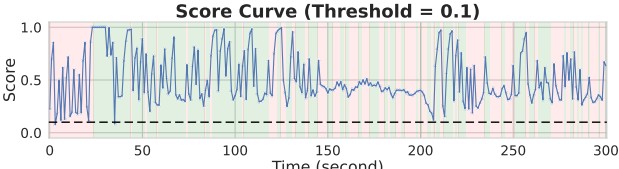

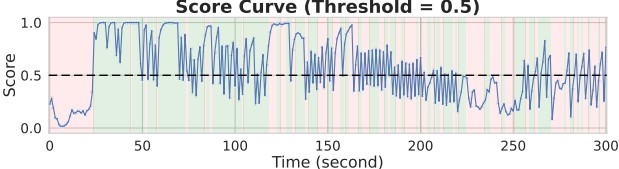

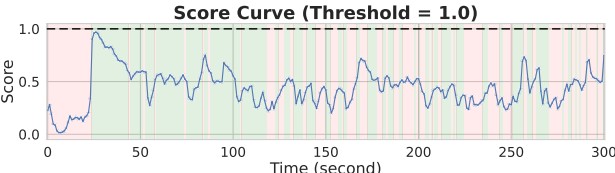

*Figure 5.* Score curve visualization. Green: labeled response; Red: labeled silence; Dashed line: threshold; Above-threshold scores: model triggers responses.

follows the label trend—response segments score slightly higher—suggesting the model can detect response-worthy moments from video alone when textual interference is absent. The middle threshold 0.5 yields the most practical pattern: an initial silent period ( 20s), followed by sustained commentary and later alternating between speaking and silence in a way that better matches real-world usage.

*Table 7.* **Inference efficiency** of Proact-VL under streaming inference. **Frame**: tokens per frame; **WS**: streaming window size (in tokens); **Mem** (GB): peak GPU memory; **Cache** (s): time to ingest the current chunk and update the KV cache; **Forward** (s): average text generation time per chunk; **Chunk** (s): end-to-end time per chunk; **Token** (s): average time per generated token.

| Frame | WS | Mem | Cache | Forward | Chunk | Token |
|---|---|---|---|---|---|---|
| 299 | 8192 | 16.07 | 0.0794 | 0.2704 | 0.3545 | 0.0433 |
| 299 | 16384 | 16.45 | 0.0850 | 0.2775 | 0.3709 | 0.0451 |
| 299 | 24576 | 16.83 | 0.0902 | 0.2728 | 0.3677 | 0.0441 |
| 299 | 32768 | 17.20 | 0.0967 | 0.3242 | 0.4313 | 0.0431 |
| 510 | 16384 | 16.57 | 0.1232 | 0.2446 | 0.3759 | 0.0454 |
| 364 | 16384 | 16.50 | 0.0975 | 0.2614 | 0.3642 | 0.0446 |

**Inference Efficiency.** As summarized in Table 7, our streaming design exhibits a clear *low-latency* profile under the streaming design. Processing each chunk mainly consists of three stages: (i) video preprocessing; (ii) ingesting the current chunk and updating the KV cache; and (iii) generating the commentary. Among them, the generation stage dominates the overall runtime, while the cache-update stage shows a mild increase as the window size and per-frame token budget grow. In contrast, the per-token generation time remains essentially constant across different settings, indicating that decoding efficiency is stable and largely insensitive to the streaming window size. Based on this predictable runtime behavior, we further estimate the real-time throughput. Assuming a fixed budget of 0.3 seconds for commentary generation and representing each frame with 364 tokens, our system is expected to stably handle 10–15 FPS video streams in practice.

### 5.7. Case Study and More Results

We provide case studies in Appendix K. In addition, we include a user study in Appendix F to evaluate Proact-VL from a human-centric perspective. The results are consistent with our automated evaluations. We further conduct general video understanding evaluations on MVBench, Video-MME, and LongVideoBench in Appendix I. The results show that our training paradigm preserves the general video understanding ability of the base model. Additional results are also provided, including, but not limited to, the robustness of the judge model in Appendix E, an ablation study on training data in Appendix G, hyperparameter analysis in Appendix J, and human alignment analysis in Appendix L.

## 6. Conclusion

In this paper, we instantiate the goal of human-like AI companions through the concrete scenarios of gaming commentary and guidance, supported by the newly proposed Live Gaming Dataset. We next introduce Proact-VL, a framework that enables real-time, proactive interaction by integrating chunk-wise processing, a lightweight response mechanism, and specialized training objectives. Extensive experiments demonstrate that Proact-VL significantly outperforms existing methods in both the quality and timing of responses.

## Impact Statement

This work presents Proact-VL, a framework for developing proactive, real-time AI companions. A primary positive impact lies in enhancing accessibility and engagement for live content, such as by providing automated, real-time commentary for esports or educational game streams, making them more informative and accessible to a wider audience. The technology could also be applied to areas like interactive education, real-time customer support, and assistive technologies. However, the development of highly responsive and human-like AI agents also necessitates careful consideration of potential risks, including the generation of misinformation or biased commentary. To address safety concerns, we have carefully cleaned the video utterance dataset used for training to ensure a healthy narrative, forming a foundational step towards responsible deployment.

## Acknowledgements

This work was supported by the National Natural Science Foundation of China under Grant No. U25B2064, the Public Welfare Research Program of Ningbo under Grant No. 2024S062, the Yongjiang Talent Project of Ningbo under Grant No. 2024A-161-G, the National Natural Science Foundation of China (Grant Nos. 62276171), and the Shenzhen Fundamental Research Project (Grant Nos. ZDCY20250901110940006, JCYJ20240813141503005).

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

# A. Model Design

## A.1. ChatML Template

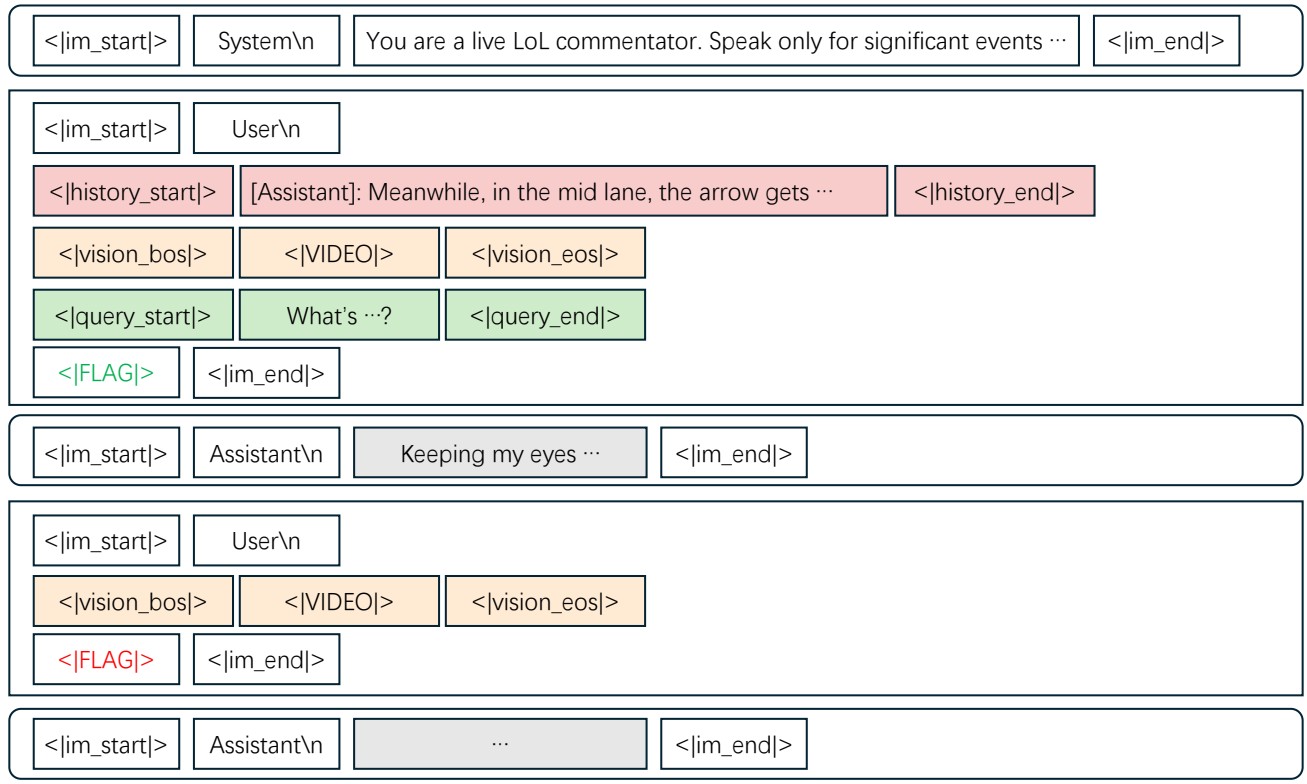

*Figure 6.* Illustration of ChatML template.

To support real-time, proactive streaming commentary, we extend the Qwen-style ChatML with a lightweight, structured input format that explicitly separates (i) environment history, (ii) the current video chunk, and (iii) the user query. Our design enables a two-stage inference pipeline—*decide-then-generate*—where the model first decides whether to respond, and only then produces a commentary clip if needed.

**Special tokens.**   We introduce five special tokens: `<|history_start|>` and `<|history_end|>` delimit the *environment history* collected from the previous timestep, including the last-second commentary produced by other assistants; `<|query_start|>` and `<|query_end|>` delimit the *user query*; and `<|FLAG|>` is a *semantic-free* marker whose hidden state is fed into a lightweight response head to compute a response score.

Importantly, we avoid reusing any existing token as the decision anchor, since common tokens inherently carry semantics and may introduce unintended priors for learning response behaviors. In contrast, `<|FLAG|>` serves as a purely structural indicator, providing a stable representation for response decision making.

**Input construction.**   Each sample consists of a system prompt followed by a single user message. The user content concatenates three parts in a fixed order: *history*, *video chunk*, and *user query*. Concretely, we format the user message as:

$$
\begin{aligned}
\text{User:} \quad & \texttt{<|history\_start|>} \; H \; \texttt{<|history\_end|>} \\
& \texttt{<|vision\_bos|>} \; \texttt{<|VIDEO|>} \; \texttt{<|vision\_eos|>} \\
& \texttt{<|query\_start|>} \; Q \; \texttt{<|query\_end|>} \\
& \texttt{<|FLAG|>},
\end{aligned}
\tag{1}
$$

where $H$ denotes the environment history (e.g., other assistants' last-second commentary), and $Q$ denotes the user query. The video chunk is represented by visual placeholders that are replaced by the corresponding streaming visual embeddings during inference.

**Decide-then-generate inference.** Given the full user content, the model first performs a *priming* forward pass and extracts the hidden state at `<|FLAG|>`. A response head then maps this hidden state to a scalar score indicating whether the model should respond at the current second. If the score exceeds a threshold, we append the `Assistant:` prefix and generate a short commentary clip; otherwise, we output a fixed silence placeholder (e.g., "`...`") to explicitly indicate no response.

**Design rationale.** Our format is motivated by three considerations.

**Ordering of history, video, and query.** We concatenate inputs as *history → video chunk → query*. The history contains last-second commentary that is not directly observable from the current video chunk. Placing it before the video allows the model to incorporate this contextual signal early, improving continuity, co-commentator coordination, and reference resolution. We place the user query after the video chunk because it is semantically closer to the assistant response; this positioning makes the query easier to attend to near generation time, strengthening user-intent conditioning.

**Why a response head instead of predicting a `<|SILENCE|>` token?** A direct token-based silence prediction is less controllable: the probability of generating `<|SILENCE|>` can vary substantially with decoding hyperparameters (e.g., temperature and top-$p$), making threshold calibration unreliable. Moreover, treating silence as a frequent assistant-side token introduces a highly imbalanced training pattern (many samples become `Assistant: <|SILENCE|>`), which can destabilize optimization and lead to degenerate collapse. In contrast, our response head produces a continuous score that is easy to threshold and decouples the response decision from text generation, yielding more stable training and controllable deployment behavior.

**Why keep the `<user><assistant>` alternation under silence?** We keep the dialogue structure consistent with Qwen-style pretraining by maintaining the `<user><assistant><user><assistant>` alternation even when the model stays silent. If we omit the assistant turn for silent timesteps, the conversation structure deviates from the base model's training distribution. Our "fill-with-silence" strategy preserves structural consistency while allowing the response head to decide whether to trigger generation.

## A.2. Infinite Inference

When the combined length of system cache, streaming cache, and current input tokens approaches the model's maximum context capacity, we implement an eviction strategy that removes the oldest 20% of the streaming cache while preserving recent interactions. This selective eviction balances context retention with memory constraints, ensuring the model maintains awareness of recent dialogue while operating within computational limits.

A critical challenge arises from positional discontinuity when evicting tokens from the middle of the sequence. To address this, we apply a lightweight reverse-RoPE (Rotary Positional Embedding) correction after each eviction operation. This technique realigns positional encodings across the cache boundary, maintaining coherent positional relationships despite the discontinuous token sequence. This efficient caching mechanism enables continuous operation over arbitrarily long streams while maintaining stable computational requirements. The system demonstrates robust performance even in extended interactive sessions, making it suitable for real-world deployment scenarios. Complete technical details of the reverse-RoPE implementation are provided in Appendix A.3.

## A.3. Reverse RoPE for windowed streaming inference.

In a streaming video assistant, the input content typically consists of three parts: a *system prompt*, *user*, and *assistant*. The system prompt defines the role, task, and identity, while user and assistant messages interleave as the video stream progresses, yielding an input pattern:

$$\langle\texttt{system}\rangle, \langle\texttt{user}\rangle, \langle\texttt{assistant}\rangle, \langle\texttt{user}\rangle, \langle\texttt{assistant}\rangle, \cdots$$

Most existing systems assign *monotonically increasing* position ids to all tokens. However, the context length of Video-LLMs is finite (e.g., Qwen-VL series typically supports up to $\sim$ 32k tokens). Over long-running inference, newly arrived user/video chunks may be encoded at position ids far beyond the frequently-seen training range, and the distance between the system prompt and recent tokens keeps growing. Empirically, this weakens instruction-following and may eventually destabilize generation. To mitigate this issue, we adopt a sliding-window mechanism: once the content length exceeds a window size $W$, we pop the oldest portion of cached tokens (e.g., the earliest 20%) and keep only recent context.

A remaining challenge is that position ids may still drift to large values over time. We address this by *re-basing* the effective

positions of the remaining cached tokens, implemented efficiently via a *reverse RoPE* operation on the KV cache. We first recall a key algebraic property of 2D rotations. Define

$$R(\theta) = \begin{pmatrix} \cos\theta & -\sin\theta \\ \sin\theta & \cos\theta \end{pmatrix}. \tag{2}$$

Then for any $a, b \in \mathbb{R}$,

$$
\begin{aligned}
R(a)R(b) &= \begin{pmatrix} \cos a & -\sin a \\ \sin a & \cos a \end{pmatrix} \begin{pmatrix} \cos b & -\sin b \\ \sin b & \cos b \end{pmatrix} \\
&= \begin{pmatrix} \cos a \cos b - \sin a \sin b & -(\cos a \sin b + \sin a \cos b) \\ \sin a \cos b + \cos a \sin b & \cos a \cos b - \sin a \sin b \end{pmatrix} \\
&= \begin{pmatrix} \cos(a+b) & -\sin(a+b) \\ \sin(a+b) & \cos(a+b) \end{pmatrix} = R(a+b),
\end{aligned} \tag{3}
$$

where we use $\cos(a+b) = \cos a \cos b - \sin a \sin b$ and $\sin(a+b) = \sin a \cos b + \cos a \sin b$. As a corollary, $R(-b) = R(b)^{-1}$ and $R(a-b) = R(a)R(-b)$.

RoPE applies such 2D rotations to each channel pair in a head. For head dimension $d$ (even), let $\{\omega_m\}_{m=0}^{\frac{d}{2}-1}$ be fixed frequencies and define the RoPE rotation at position $p$ as the block-diagonal matrix

$$\mathcal{R}(p) = \mathrm{diag}\big(R(p\omega_0), R(p\omega_1), \ldots, R(p\omega_{\frac{d}{2}-1})\big). \tag{4}$$

By applying (3) to each block, we have the block-wise additivity

$$\mathcal{R}(p_1)\mathcal{R}(p_2) = \mathcal{R}(p_1 + p_2), \qquad \mathcal{R}(p - \Delta) = \mathcal{R}(-\Delta)\mathcal{R}(p). \tag{5}$$

Let a cached key at position $p$ be RoPE-encoded as $k_p^{\mathrm{rope}} = \mathcal{R}(p)k_p$. Then applying a reverse rotation yields an *exact* position shift:

$$\hat{k}_p^{\mathrm{rope}} = \mathcal{R}(-\Delta)\, k_p^{\mathrm{rope}} = \mathcal{R}(-\Delta)\mathcal{R}(p)k_p = \mathcal{R}(p - \Delta)k_p, \tag{6}$$

which is equivalent to recomputing RoPE using the shifted position id $p' = p - \Delta$ without re-encoding past tokens.

**Applying reverse RoPE to KV cache.** Concretely, suppose the key cache corresponds to position ids $\{p_0, \ldots, p_I\}$, where $\{p_0, \ldots, p_{i-1}\}$ belong to the system prompt and $\{p_i, \ldots, p_I\}$ come from interleaved user/assistant chunks. When the window overflows, we pop the first $j$ cached tokens and keep $\{p_j, \ldots, p_I\}$. To re-base the remaining cache into a stable coordinate system, we set $\Delta = p_j$ and apply (6) to all remaining cached keys:

$$k^{\mathrm{rope}}(p) \leftarrow \mathcal{R}\big(-(p_j - p_i)\big)\, k^{\mathrm{rope}}(p) = \mathcal{R}(p_i - p_j)\, k^{\mathrm{rope}}(p), \qquad \forall p \in \{p_j, \ldots, p_I\}. \tag{7}$$

so their effective position ids become $p' = p - p_j + p_i$ (i.e., starting from $i$). For subsequent decoding, we assign the new query position id in the same re-based coordinate system (e.g., $p_{\mathrm{new}} = \max(p'_{\mathrm{cache}}) + 1$), ensuring consistency between queries and cached keys.

# B. Data Construction

### B.1. Data Preprocessing

Our raw data consist of videos paired with ASR transcripts, where each ASR segment is associated with a time interval and its corresponding commentary caption. To obtain streaming-friendly, per-second supervision, we convert each segment-level caption into a sequence of second-level captions.

Given an ASR segment with start and end timestamps, we first round both boundaries to integer seconds. Suppose the segment spans $t$ seconds and its caption contains $n$ words. We distribute the words into $t$ one-second bins as evenly as possible. Let

$$q = \left\lfloor \frac{n}{t} \right\rfloor, \qquad r = n - tq.$$

Then the first $r$ seconds are assigned $q + 1$ words each, and the remaining $t - r$ seconds are assigned $q$ words each, yielding a per-second caption sequence $\{c_s, c_{s+1}, \ldots, c_{s+t-1}\}$ aligned to the rounded timeline.

To indicate that an utterance is still ongoing within a segment, we append an ellipsis suffix " ..." to the captions of the first $t - 1$ seconds (i.e., $c_s$ to $c_{s+t-2}$), while keeping the last second $c_{s+t-1}$ unchanged.

### B.2. Benchmark Construction

Based on the constructed dataset, we split a training set and two complementary test sets: *Live Gaming Benchmark* and *Live Gaming Benchmark-Streaming*. The training set spans ten games, including *Cyberpunk 2077*, *StarCraft II*, *Baldur's Gate 3*, *Elden Ring*, *Tears of the Kingdom*, *Yu-Gi-Oh*, *League of Legends*, *CSGO*, *Street Fighter 6*, and *Minecraft*. We perform a video-wise split for each game, using $80\%$ of videos for training, $10\%$ for testing, and reserving the remaining $10\%$ for future use. For training data, we segment each video into 36-second clips with an 18-second overlap between adjacent clips. For *Minecraft*, we additionally create 60-second clips as an extra slicing variant.

**Live Gaming Benchmark.**    This benchmark evaluates clip-level performance for both commentary and guidance scenarios, and additionally includes an in-domain generalization game, *Black Myth: Wukong*. Motivated by the observation that desirable response density typically falls within a moderate response-rate range (roughly $30\%$–$70\%$), we stratify clips by response rate and sample per game: 60 clips from the 0–$30\%$ bin, 120 clips from the 30–$70\%$ bin, and 60 clips from the 70–$100\%$ bin. For *Minecraft*, we sample from both 36-second and 60-second clips. Overall, the benchmark contains 2,640 in-distribution test clips and 240 additional test clips from a different game distribution.

**Live Gaming Benchmark-Streaming.**    To evaluate long-horizon, continuous commentary, we construct a streaming test set by selecting full-length videos rather than short clips. Specifically, from both *Solo Commentary* and *Co-Commentary* settings, we choose one complete video per game for evaluation. The resulting set includes one 30-minute video, eight 1-hour videos, and one 2-hour video, enabling assessment of sustained commentary quality and stability over extended streams.

### B.3. Training and Test Data Distribution

Table 8 summarizes the data distribution across splits. Our training set contains 128,000 samples drawn from 12 sources (10 game domains plus two general streaming datasets, LiveCC and Ego4D). For evaluation, we use a game-centric test set with 9 games sampled uniformly (240 clips per game, 2,160 clips in total), plus *Minecraft* with two clip-length variants (240 clips with length 36s and 240 clips with length 60s, 480 clips total), and an *Ego4D* subset of 134 samples consisting of 300-second videos. In addition, we include 240 clips from an in-domain generalization game, *Black Myth: Wukong*.

# C. Experiment

### C.1. Evaluation Metrics

**TimeDiff Metric**    We define *TimeDiff* to evaluate the temporal accuracy of predicted responses with respect to ground-truth annotations. Let the $i$-th ground-truth interval be denoted as $[a_i, b_i]$, with ground-truth onset time $t_i^{\text{gt}} = a_i$. Predicted responses are indexed by $k$ and characterized by their start times $\hat{t}_k$.

*Table 8.* Data distribution across splits.

| Source | Train | Test |
|---|---|---|
| Baldur's Gate 3 | 10,286 | 240 |
| CS:GO | 10,347 | 240 |
| Cyberpunk 2077 | 7,986 | 240 |
| Elden Ring | 7,398 | 240 |
| League of Legends | 5,506 | 240 |
| Minecraft | 6,543 | 480 |
| StarCraft II | 11,827 | 240 |
| Street Fighter 6 | 10,000 | 240 |
| Tears of the Kingdom | 10,077 | 240 |
| Yu-Gi-Oh | 3,183 | 240 |
| LiveCC | 31,991 | - |
| Ego4D | 12,856 | 134 |
| Black Myth: Wukong | - | 240 |
| Total | 128,000 | 3,014 |

We first define the original TimeDiff for the $i$-th ground-truth interval as

$$\text{TimeDiff}_{\text{orig}}(i) = \begin{cases} \min\limits_{k \in \mathcal{K}_i} \left| \hat{t}_k - t_i^{\text{gt}} \right|, & \text{if } \mathcal{K}_i \neq \emptyset, \\ b_i - a_i, & \text{otherwise,} \end{cases} \tag{8}$$

where $\mathcal{K}_i = \{k \mid \hat{t}_k \in [a_i, b_i]\}$ denotes the set of predicted responses whose start times fall within the $i$-th ground-truth interval.

To penalize redundant or misaligned predictions outside a temporal tolerance window, we define the expanded interval $\Delta_i = [a_i - \delta, b_i + \delta]$. The refined *TimeDiff* metric is then formulated as

$$\text{TimeDiff}(i) = \text{TimeDiff}_{\text{orig}}(i) + \alpha \sum_{k \in \mathcal{P}_i} \mathbb{I}\left[ \hat{t}_k \notin \Delta_i \right], \tag{9}$$

where $\delta$ denotes the tolerance threshold, $\alpha$ is a penalty coefficient, and $\mathcal{P}_i$ represents the set of predictions associated with the $i$-th ground-truth interval.

**PAUC Metric** PAUC is a specialized metric for evaluating proactive interaction models, inspired by user journey mapping and formulated as a dynamic trajectory over a video sequence. By integrating response quality along the temporal axis, PAUC captures the cumulative impact and temporal dynamics of proactive behaviors rather than treating responses as isolated events. We use GPT-5.1 as the judge and additionally report results obtained with an initial score of 0 and $\omega = 0.5$.

---

**System Prompt for PAUC**

You are an evaluator for a gaming commentary system. Your task is to rate the whether the predicted answer covers the key points of the ground truth answer. Use the following scale to assign a score:
- 3: Mostly covered; the predicted answer covers all key information in the ground truth answer, though it may have minor inaccuracies or rephrases.
- 2: Partially covered; the predicted answer has some correct information, but also contains significant inaccuracies or missing key points.
- 1: Incorrect; the predicted answer may be related to the ground truth answer, but most of the information is missing, or the predicted answer is of very poor quality.
The similarity between past and current Predicted Answers must be considered, and penalties will be applied to both cases.
Output the score only, do not add more explanations.

---

**F1 Metric** Existing evaluation metrics exhibit complementary yet critical limitations in assessing proactive behaviors. The *TimeDiff* metric only considers the best-aligned model response within each ground-truth window, thereby failing to

characterize model behavior over the full temporal extent of the video. In contrast, *PAUC* evaluates responses exclusively within ground-truth intervals, ignoring predictions outside these intervals and providing limited penalization for over-proactive behavior. As a result, neither metric jointly captures temporal recall and false positives along the entire timeline.

To address these limitations, we model the entire video as a temporal axis and define a binary indicator function over time, treating timestamps within ground-truth intervals as positive regions and all remaining timestamps as negative regions. Precision and recall are then computed by comparing the model's response timeline against this temporal labeling, from which the F1 score is derived. This formulation enables a holistic evaluation of proactive performance, jointly accounting for timely triggering and spurious responses, and thus providing a clearer characterization of a model's proactive capability.

## C.2. LLM Judge Score

We evaluate commentary from two complementary perspectives: **LiveU** for second-level streaming usability and **FinalQ** for the consolidated script quality after concatenation (timestamps removed; silent seconds ignored).

**Time**: whether the model speaks at the right moments (coverage of salient windows and, in multi-commentator mode, minimal disruptive overlap).
**Rate**: whether the pacing is listenable in real time (brief, non-dumpy bursts; step-by-step in guidance).
**TextU**: whether each spoken unit is immediately clear as live speech.
**LiveU** is the mean of **Time**, **Rate**, and **TextU**.

**Fidelity**: whether the final script avoids direct event contradictions and misleading concrete claims.
**Continuity**: whether it stays coherent with the provided context and thread.
**Substance**: whether it is useful, readable, and non-redundant.
**FinalQ** is the mean of **Fidelity**, **Continuity**, and **Substance**.

## C.3. Experimental Detail

**Offline Labels.** For GPT-series model, we use GPT-4o-2024-11-20 to generate offline captions and offline streaming commentary. For each video, frames are sampled at 1 FPS. The visual input is constrained with max_pixels = 384 × 28 × 28 and min_pixels = 128 × 28 × 28 for each channel. For generating offline caption labels, since the commercial API supports at most 50 images per request, videos exceeding this limit are split into multiple chunks (each containing up to 50 frames). Captions are generated for each chunk independently, and finally merged in chronological order to form the complete caption for the entire video.

**Training.** We fine-tune our model from different backbones, including Qwen2-VL, Qwen2.5-VL, Qwen3-VL, and LiveCC-Base. Except for the choice of base model, we keep all training hyperparameters identical across runs to ensure fair comparisons.

For the input resolution constraints, we follow a unified pixel budgeting strategy for visual inputs with `MIN_PIXELS` = $128 \times 28 \times 28$ and `MAX_PIXELS` = $540 \times 28 \times 28$. For videos, we additionally cap the total pixels by `MAX_VIDEO_PIXELS` = $36 \times 540 \times 28 \times 28$. Given a video chunk of duration $T$ (in seconds), the per-frame pixel budget is set to $\min(\texttt{MAX\_PIXELS}, \texttt{MAX\_VIDEO\_PIXELS}/T)$, and we decode videos at 2 FPS.

The system prompt is composed of three parts: *role*, *persona*, and *task*. The role specifies the game-specific commentator identity (e.g., "You are a live commentator for a Cyberpunk 2077 game."), with one fixed role template per game. The persona is mined from our data collection pipeline. The task introduces the downstream setting (Solo Commentary, Co-Commentary, or Guidance); we prepare six task templates in total. During training, we concatenate the game-specific role prompt, one randomly sampled persona, and one randomly sampled task template. This randomization improves robustness to system-prompt variations and enhances generalization to diverse application scenarios.

For the main causal language modeling loss $\mathcal{L}_{\text{main}}$, we compute loss only on tokens generated by the *active* assistant, up to `<|im_end|>`, and mask out all tokens corresponding to silent assistants. For the response loss $\mathcal{L}_{\text{resp}}$, supervision is applied only at the `<|FLAG|>` token position, using the per-step speak/silence label.

## C.4. Results from Training with Different Base Models

*Table 9.* Comparison of downstream task performance of Proact-VL models trained with different base models.

| Model | Solo Commentary | | Co-Commentary | | Guidance | | Ego4d | | Black Myth Wukong | | Overall | |
|---|---|---|---|---|---|---|---|---|---|---|---|---|
| | CC ↑ | F1 ↑ | CC ↑ | F1 ↑ | CC ↑ | F1 ↑ | CC ↑ | F1 ↑ | CC ↑ | F1 ↑ | CC ↑ | F1 ↑ |
| Qwen2-VL | 28.67 | 22.41 | 38.44 | 38.00 | 27.08 | 24.62 | 6.75 | 16.36 | 55.62 | 17.47 | 31.31 | 23.77 |
| **Proact-VL**$_{\text{Qwen2-VL}}$ ($\tau = 0.5$) | 53.79 | 56.21 | 54.90 | 70.09 | 40.94 | 50.68 | 61.94 | 46.72 | 64.38 | 46.30 | 55.19 | 54.00 |
| Qwen2.5-VL | 12.75 | 62.31 | 20.26 | 61.53 | 23.44 | 45.21 | 3.17 | 16.36 | 21.67 | 57.53 | 16.26 | 48.59 |
| **Proact-VL**$_{\text{Qwen2.5-VL}}$ ($\tau = 0.3$) | 49.62 | 63.66 | 50.94 | 75.84 | 42.71 | 52.94 | 58.58 | 44.72 | 56.25 | 59.68 | 51.62 | 59.37 |
| Qwen3-VL | 15.04 | 51.85 | 40.94 | 24.34 | 35.62 | 36.91 | 19.05 | 16.36 | 14.79 | 49.51 | 25.09 | 35.79 |
| **Proact-VL**$_{\text{Qwen3-VL}}$ ($\tau = 0.3$) | 48.79 | 64.51 | 51.82 | 76.47 | 47.50 | 53.17 | 68.66 | 48.00 | 55.21 | 60.43 | 54.40 | 60.52 |

# D. Game-Wise Analysis

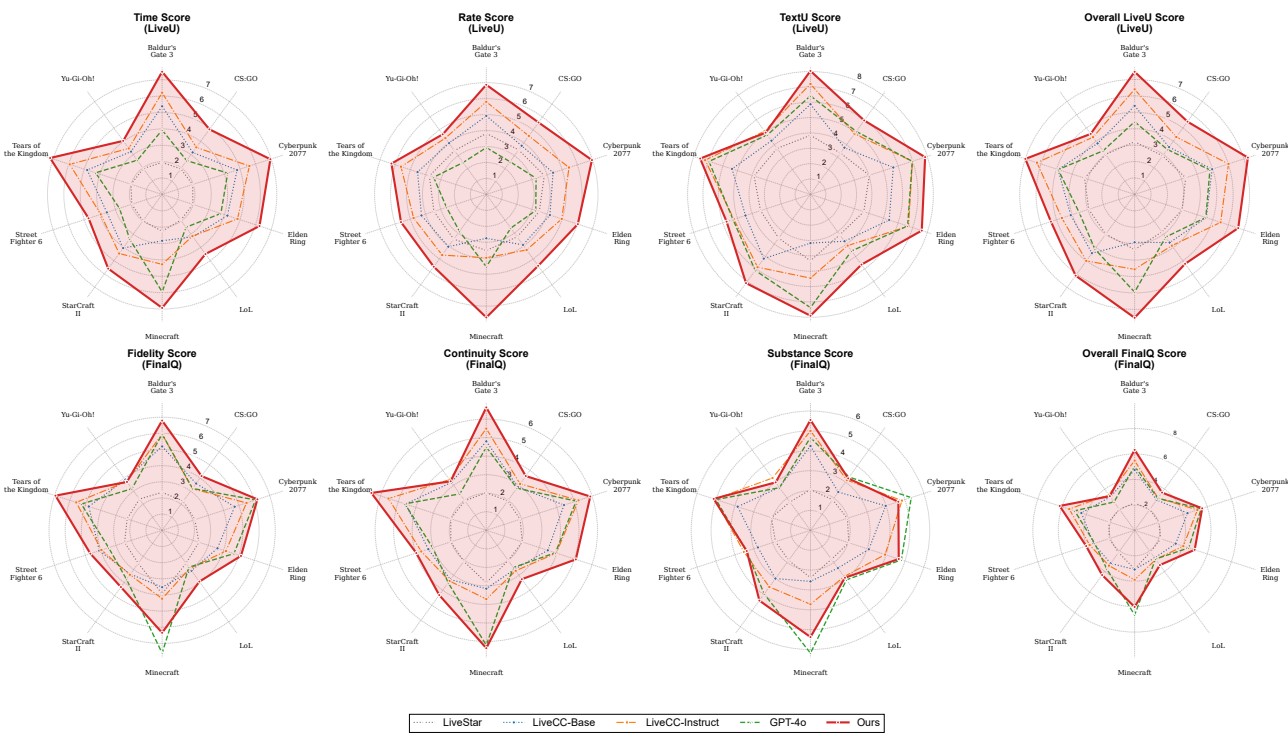

*Figure 7.* Game-Wise Analysis.

We further break down performance by game to examine whether the gains of Proact-VL are consistent across diverse gameplay styles and visual contexts. Fig. 7 summarizes the results using radar plots, where each axis corresponds to a game and each polygon corresponds to a model. We report both all the llm judge score, Time, Rate, TextU, LiveU for response quality, and Fidelity, continuity, substance and FianlQ for text quality.

Overall, Proact-VL exhibits clear and consistent improvements across games. In particular, our method achieves higher LiveU and FinalQ scores on almost all games, indicating that the proposed proactive response mechanism and clip-level generation strategy generalize well beyond a single title. The gains are especially pronounced in the overall scores, suggesting that Proact-VL improves both streaming usability and the quality of the concatenated commentary.

# E. Robustness of LLM-as-a-Judge Evaluation

*Table 10.* Robustness of LLM-as-a-Judge on SOLO commentary scenario.

| Caption Model | Judge Model | Model | Time | Rate | TextU | Fidelity | Continuity | Substance | LiveU | FinalQ | Win Rate |
|---|---|---|---|---|---|---|---|---|---|---|---|
| Gemini 2.5 Pro | GPT-5.1 | LiveCC-7B-Base | 4.65 | 4.42 | 5.49 | 4.33 | 4.11 | 3.63 | 4.85 | 4.02 | 41.17 |
| Gemini 2.5 Pro | GPT-5.1 | LiveCC-7B-Instruct | 5.41 | 5.31 | 6.79 | 4.91 | 4.72 | 4.47 | 5.84 | 4.70 | 34.33 |
| Gemini 2.5 Pro | GPT-5.1 | Proact-VL | 6.70 | 6.34 | 7.64 | 5.88 | 5.69 | 4.87 | 6.89 | 5.48 | 53.62 |
| GPT-4o | Gemini 2.5 Flash | LiveCC-7B-Base | 6.71 | 6.77 | 6.99 | 5.63 | 5.23 | 5.42 | 6.82 | 5.43 | 64.83 |
| GPT-4o | Gemini 2.5 Flash | LiveCC-7B-Instruct | 7.21 | 7.26 | 7.56 | 5.66 | 5.06 | 5.50 | 7.34 | 5.41 | 68.75 |
| GPT-4o | Gemini 2.5 Flash | Proact-VL | 7.41 | 7.47 | 7.71 | 6.30 | 5.74 | 5.91 | 7.53 | 5.98 | 76.79 |

*Table 11.* Robustness of LLM-as-a-Judge on co-commentary scenario.

| Caption Model | Judge Model | Model | Time | Rate | TextU | Fidelity | Continuity | Substance | LiveU | FinalQ | Win Rate |
|---|---|---|---|---|---|---|---|---|---|---|---|
| Gemini 2.5 Pro | GPT-5.1 | LiveCC-7B-Base | 3.29 | 3.98 | 4.08 | 3.66 | 2.99 | 2.55 | 3.78 | 3.07 | 45.89 |
| Gemini 2.5 Pro | GPT-5.1 | LiveCC-7B-Instruct | 3.53 | 4.50 | 4.83 | 3.44 | 3.22 | 3.19 | 4.29 | 3.28 | 37.40 |
| Gemini 2.5 Pro | GPT-5.1 | Proact-VL | 4.54 | 5.34 | 5.58 | 4.11 | 3.53 | 3.13 | 5.15 | 3.59 | 51.46 |
| GPT-4o | Gemini 2.5 Flash | LiveCC-7B-Base | 7.02 | 7.09 | 7.26 | 5.39 | 4.99 | 4.99 | 7.12 | 5.12 | 51.93 |
| GPT-4o | Gemini 2.5 Flash | LiveCC-7B-Instruct | 7.40 | 7.37 | 7.65 | 5.17 | 4.93 | 5.03 | 7.47 | 5.04 | 54.11 |
| GPT-4o | Gemini 2.5 Flash | Proact-VL | 7.45 | 7.50 | 7.68 | 5.59 | 5.28 | 5.28 | 7.54 | 5.38 | 61.72 |

*Table 12.* Robustness of LLM-as-a-Judge on guidance scenario.

| Caption Model | Judge Model | Model | Time | Rate | TextU | Fidelity | Continuity | Substance | LiveU | FinalQ | Win Rate |
|---|---|---|---|---|---|---|---|---|---|---|---|
| Gemini 2.5 Pro | GPT-5.1 | LiveCC-7B-Base | 2.83 | 2.75 | 3.17 | 3.52 | 3.15 | 2.57 | 2.92 | 3.08 | 29.58 |
| Gemini 2.5 Pro | GPT-5.1 | LiveCC-7B-Instruct | 4.27 | 3.97 | 5.45 | 4.23 | 3.71 | 3.72 | 4.56 | 3.89 | 13.33 |
| Gemini 2.5 Pro | GPT-5.1 | Proact-VL | 6.93 | 7.71 | 7.90 | 6.34 | 6.36 | 5.37 | 7.51 | 6.02 | 42.60 |
| GPT-4o | Gemini 2.5 Flash | LiveCC-7B-Base | 6.52 | 6.79 | 7.03 | 4.75 | 4.35 | 4.45 | 6.78 | 4.52 | 22.92 |
| GPT-4o | Gemini 2.5 Flash | LiveCC-7B-Instruct | 4.31 | 4.46 | 5.06 | 3.56 | 2.74 | 3.25 | 4.61 | 3.18 | 33.75 |
| GPT-4o | Gemini 2.5 Flash | Proact-VL | 7.15 | 7.43 | 7.57 | 6.17 | 5.81 | 5.94 | 7.38 | 5.97 | 51.46 |

**Robustness to judge models.** To assess the robustness of LLM-as-a-Judge evaluation, we replace the judge model. Tab. 10, 11, and 12 report results on SOLO, CO, and GUIDANCE. Although absolute scores vary due to different calibration across judges, the high-level conclusion remains consistent: **Proact-VL** achieves the best overall performance under both judges, ranking first in *LiveU* and *FinalQ* across all three settings, and also obtaining the highest win rate. We observe minor metric-level re-orderings among the baselines in a few cases (e.g., some dimensions under GUIDANCE), but the main claim that Proact-VL outperforms prior LiveCC baselines is stable to the choice of judge model.

*Table 13.* Judge stochasticity over 5 runs for Proact-VL.

| Run | Solo (%) | Co (%) | Guidance (%) |
|---|---|---|---|
| 1 | 53.62 | 51.46 | 42.60 |
| 2 | 53.79 | 51.72 | 42.50 |
| 3 | 53.67 | 51.25 | 43.02 |
| 4 | 53.92 | 51.35 | 43.23 |
| 5 | 53.75 | 51.72 | 43.23 |

*Table 14.* Summary statistics of judge stochasticity (5 runs).

| Setting | Mean (%) | Std (%) | 95% CI (%) |
|---|---|---|---|
| Solo | 53.75 | 0.12 | [53.65, 53.85] |
| Co | 51.50 | 0.21 | [51.31, 51.69] |
| Guidance | 42.92 | 0.35 | [42.61, 43.22] |

**Robustness to judge stochasticity.** We examine the stability of LLM-as-a-Judge evaluation by repeating the judging process five times for Proact-VL under response threshold $= 0.3$. Tab. 13 lists the per-run win rates on SOLO, CO, and GUIDANCE, while Tab. 14 summarizes the mean, standard deviation, and 95% confidence intervals. Across all three settings, the variability is small, indicating that our reported win-rate results are stable under repeated LLM-judge runs.

# F. User Study

**Baselines and Data Collection.** To comprehensively evaluate the performance of our model, we conducted an extensive user study. Regarding the selection of baselines, we chose one representative model from each of three distinct categories: Offline,

*Table 15.* Performance comparison across different games.

| Model | Black Myth Wukong | CSGO | Minecraft |
|---|---|---|---|
| Gemini 2.5 Pro | 20.0 | 20.0 | 0.0 |
| Livestar | 10.0 | 10.0 | 0.0 |
| LiveCC-7B-Instruct | 30.0 | 10.0 | 10.0 |
| **Proact-VL** | **80.0** | **86.7** | **96.7** |

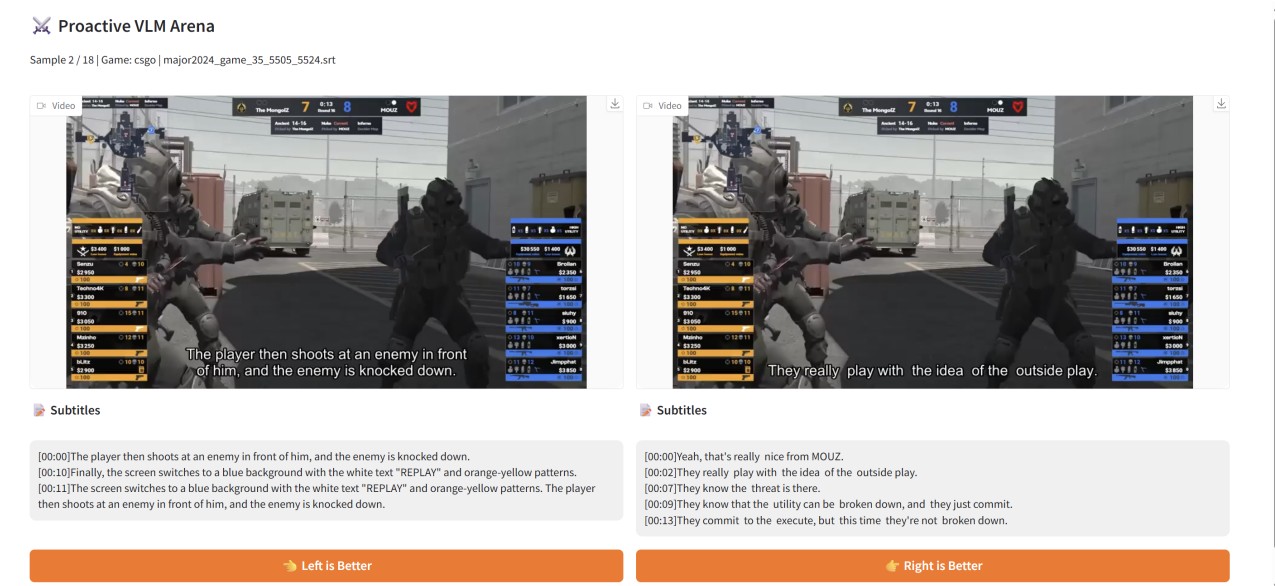

*Figure 8.* Interface for Pairwise Model Comparison.

Proactive, and Real-Time models. The experimental data covers three scenarios: single-person commentary, multi-person commentary, and game instruction. For each scenario, we selected a typical game and extracted 10 video clips, resulting in a total of 30 test samples.

**Evaluation Platform and Stimuli Generation.** We constructed a specialized evaluation platform to conduct the testing, the interface of which is illustrated in Figure 8. To provide evaluators with a more intuitive audiovisual experience, we rendered the model's textual output as synchronized subtitles and utilized Text-to-Speech (TTS) technology to synthesize speech, which was then embedded directly into the video clips.

**User Study Protocol and Results.** The evaluation adopted a pairwise comparison method. Specifically, each comparison pair consisted of our proposed model and one of the three baseline models. Evaluators were required to determine the superior performance between the two after viewing the videos. To mitigate subjective bias and ensure result reliability, each pair was independently assessed by three different evaluators to establish inter-annotator reliability. We recruited 15 evaluators for this study, with each participant randomly assigned to complete 18 evaluation tasks. Table 15 presents the Win Rate of our model against the baselines across different game scenarios. The results show that, in the user study, our model significantly outperforms the highly competitive models from all three categories (Offline Models, Proactive Models, and Real-Time Models).

## G. Ablation Study for Training Data

We conduct a data ablation study by removing one training source at a time while keeping the experimental setup identical to the full model, including 2000 training steps. Table 16 shows that Proact-VL achieves the best overall performance when trained on the full mixture. Each data source contributes substantially to its corresponding domain. Without Gaming data, Gaming CC drops by 13.08%. Without Ego4D data, Ego4D performance decreases markedly with a 22.39% drop. Without Live-SFT, Livesports CC drops by 7.69%. Importantly, after fusing all three sources, the model maintains strong

*Table 16.* Effectiveness of Training Data.

| Model | Ego4D | | Gaming | | Livesports |
|---|---|---|---|---|---|
| | CC | F1 | CC | F1 | CC |
| LiveCC-7B-Base | 12.69 | 14.94 | 38.88 | 36.10 | 70.04 |
| Proact-VL(w/o game) | 56.34 | 47.13 | 37.83 | 50.93 | 78.88 |
| Proact-VL(w/o ego4d) | 36.19 | 33.08 | 51.84 | 61.21 | 74.82 |
| Proact-VL(w/o live-sft) | 57.46 | 45.77 | 51.56 | 60.50 | 64.78 |
| Proact-VL | 58.58 | 47.95 | 50.91 | 60.08 | 72.47 |

generalization. The Gaming score only exhibits a small decrease compared to the best in-domain variant, and the Livesports decrease remains acceptable while still outperforming the base model. Notably, the full model reaches the strongest results on Ego4D, which we attribute to guidance-style supervision also present in the Gaming data, providing transferable signals that further improve egocentric narration.

## H. Effectiveness of Prompts

Our framework injects prompts at two complementary stages. First, we augment the *system prompt* with task instructions and a lightweight persona, encouraging the model to follow the desired commentary style and tone. Second, we structure the *user template* to include step-wise context, namely the previous (history) content and an optional user query, so the model can ground each chunk-level response in recent dialogue and interactions. In this section, we study how such prompt injections affect baseline models.

*Table 17.* Prompt injection ablation on LiveCC-Base and LiveCC-Instruct. "system" indicates injecting task/persona instructions into the system prompt; "user" indicates injecting history/query context into the user template.

| Prompt | | Solo | | Co | | Guidance | |
|---|---|---|---|---|---|---|---|
| system | user | CC | F1 | CC | F1 | CC | F1 |
| *LiveCC-Base* | | | | | | | |
| - | - | 41.46 | 47.05 | 54.79 | 29.71 | 30.73 | 12.08 |
| ✓ | ✓ | 44.71 | 16.40 | 45.78 | 30.62 | 35.52 | 16.77 |
| - | ✓ | 41.17 | 47.05 | 45.89 | 43.25 | 29.58 | 18.00 |
| *LiveCC-Instruct* | | | | | | | |
| - | - | 34.67 | 62.41 | 35.26 | 61.27 | 11.98 | 44.48 |
| ✓ | ✓ | 35.96 | 61.23 | 36.72 | 60.82 | 13.96 | 44.92 |
| - | ✓ | 34.33 | 62.05 | 37.40 | 61.26 | 13.33 | 44.83 |

Table 17 shows that prompt injection has a non-trivial impact on both speaking behavior (F1) and commentary consistency (CC), and the effect depends on the base model type. For the pretrained *LiveCC-Base*, adding full prompt constraints (system+user) substantially alters the response behavior: while CC can increase in Solo/Guidance, the F1 in Solo drops sharply (47.05→16.40), indicating that overly strong instructions can over-constrain the model and suppress timely responses. In addition, injecting only user-side context (history/query) can already change the trade-off notably (e.g., Co F1 improves from 29.71 to 43.25), but may also reduce CC in Co, suggesting sensitivity to how the context is framed.

For *LiveCC-Instruct*, prompt injection yields relatively modest changes: both CC and F1 vary only slightly across settings, implying that instruction-tuned models are more robust to prompt formatting, but still benefit from lightweight contextual grounding in the user template.

Overall, overly heavy prompt constraints can significantly degrade model performance (especially for base models), while removing prompts entirely makes the model less aware of the intended task and interaction format. Therefore, in our main experiments we adopt a minimally invasive prompting strategy: we keep the system prompt as close as possible to the original system prompt of the backbone model, and inject only the essential information (user query and background/history content) in a compact user template.

*Table 18.* Per-task accuracy (%) comparison with a Qwen3-VL baseline on MVBench. Δ denotes Proact-VL−Base.

| Task | Base | Proact-VL | Δ |
|---|---|---|---|
| **Action** | | | |
| Action Sequence | 73.0 | **78.0** | +5.0 |
| Action Prediction | 64.0 | **64.5** | +0.5 |
| Action Antonym | 82.5 | **83.0** | +0.5 |
| Fine-grained Action | 46.0 | **47.0** | +1.0 |
| Unexpected Action | **80.0** | **80.0** | +0.0 |
| Action Localization | 51.5 | **52.0** | +0.5 |
| Action Count | **41.0** | 37.0 | -4.0 |
| **Object** | | | |
| Object Existence | **88.5** | 81.0 | -7.5 |
| Object Interaction | 69.5 | **74.5** | +5.0 |
| Object Shuffle | **40.0** | 37.0 | -3.0 |

| Task | Base | Proact-VL | Δ |
|---|---|---|---|
| **Motion** | | | |
| Moving Direction | **60.5** | 54.5 | -6.0 |
| Moving Count | **70.5** | 62.5 | -8.0 |
| Moving Attribute | **93.0** | 81.0 | -12.0 |
| **State & Reasoning** | | | |
| Scene Transition | **89.0** | 84.0 | -5.0 |
| State Change | 74.5 | **76.0** | +1.5 |
| Fine-grained Pose | 71.5 | **75.0** | +3.5 |
| Character Order | 73.0 | **73.5** | +0.5 |
| Egocentric Navigation | 38.5 | **39.0** | +0.5 |
| Episodic Reasoning | **54.5** | **54.5** | +0.0 |
| Counterfactual Inference | **65.0** | 59.0 | -6.0 |
| **Overall** | **66.3** | 64.7 | -1.6 |

*Table 19.* Results on Video-MME using 8 frames. "Subs" indicates whether subtitles are used.

| Model | Subs | Short | Medium | Long | Overall |
|---|---|---|---|---|---|
| Qwen3-VL | – | 66.4 | 55.4 | 49.9 | 57.3 |
| Proact-VL$_{\text{Qwen3-VL}}$ | – | **67.6** | **58.2** | **51.4** | **59.1** |
| Qwen3-VL | ✓ | 68.0 | 56.7 | 50.3 | 58.3 |
| Proact-VL$_{\text{Qwen3-VL}}$ | ✓ | **68.9** | **57.4** | **53.1** | **59.8** |
| Qwen2.5-VL | – | 61.9 | 51.2 | 46.2 | 53.1 |
| Proact-VL$_{\text{Qwen2.5-VL}}$ | – | **64.8** | **56.8** | **49.7** | **57.1** |
| Qwen2.5-VL | ✓ | 64.0 | 52.8 | 47.6 | 54.8 |
| Proact-VL$_{\text{Qwen2.5-VL}}$ | ✓ | **67.1** | **57.4** | **50.8** | **58.4** |

# I. General Video Understanding Evaluation

To examine whether proactive fine-tuning degrades general video understanding ability, we conduct offline evaluations on three widely used video understanding benchmarks: MVBench (Li et al., 2024), Video-MME (Fu et al., 2025), and LongVideoBench (Wu et al., 2024). These benchmarks cover complementary aspects of video comprehension, including action and temporal understanding, object interaction, subtitle-assisted video question answering, and long-context video reasoning. Since our training primarily targets vertical-domain gaming data and proactive assistance objectives, these evaluations serve as a stress test for whether Proact-VL preserves the base model's general-purpose video understanding capability. All these experiments are conducted using VLMEvalKit (Duan et al., 2024).

**MVBench.** As shown in Table 18, Proact-VL attains a comparable aggregate accuracy to the Qwen3-VL baseline (64.7% vs. 66.30%, $\Delta = -1.6$). The model improves on several action-centric and interaction-related tasks, such as *Action Sequence* (+5.0) and *Object Interaction* (+5.0), while showing regressions on some motion- and attribute-heavy tasks, such as *Moving Attribute* (-12.0), *Moving Count* (-8.0), and *Object Existence* (-7.5). This suggests that proactive fine-tuning does not cause severe degradation on MVBench, but introduces task-level fluctuations depending on the type of visual reasoning required.

**Video-MME.** We further evaluate Proact-VL on Video-MME using 8 uniformly sampled frames, under both subtitle-free and subtitle-assisted settings. As shown in Table 19, Proact-VL consistently improves over the corresponding base models. For Qwen3-VL, Proact-VL improves the overall score from 57.3 to 59.1 without subtitles and from 58.3 to 59.8 with subtitles. For Qwen2.5-VL, the gain is larger, improving the overall score from 53.1 to 57.1 without subtitles and from 54.8 to 58.4 with subtitles. These results indicate that proactive fine-tuning preserves, and in this case even improves, general video question-answering performance on Video-MME.

**LongVideoBench.** Finally, we evaluate Proact-VL on LongVideoBench using 8 frames to examine long-video understanding under a constrained visual budget. As shown in Table 20, Proact-VL shows mixed but limited changes across

*Table 20.* Results on LongVideoBench using 8 frames. Columns denote different video duration ranges in seconds.

| Model | 15 | 60 | 600 | 3600 | Overall |
|---|---|---|---|---|---|
| Qwen3-VL | **70.4** | 64.0 | **55.1** | **45.7** | **54.5** |
| Proact-VL$_{Qwen3-VL}$ | 66.7 | **68.6** | 51.0 | 44.7 | 52.8 |

different video duration ranges. It improves on the 60-second split, increasing the score from 64.0 to 68.6, while slightly decreasing on the 15-second, 600-second, and 3600-second splits. Overall, the score changes from 54.5 to 52.8. This suggests that proactive fine-tuning may introduce some benchmark-specific trade-offs for long-video reasoning, but the degradation remains modest.

**Overall Analysis.** Across MVBench, Video-MME, and LongVideoBench, Proact-VL exhibits small fluctuations rather than a systematic degradation in general video understanding ability. It achieves clear gains on Video-MME, remains comparable on MVBench, and shows a modest decrease on LongVideoBench. Since our training primarily targets vertical-domain gaming data and proactive assistance objectives rather than generic offline video understanding, these results suggest that Proact-VL largely preserves the base model's general-purpose video comprehension capability while acquiring proactive domain-specific behavior.

# J. Hyperparameter Analysis

## J.1. Response Threshold

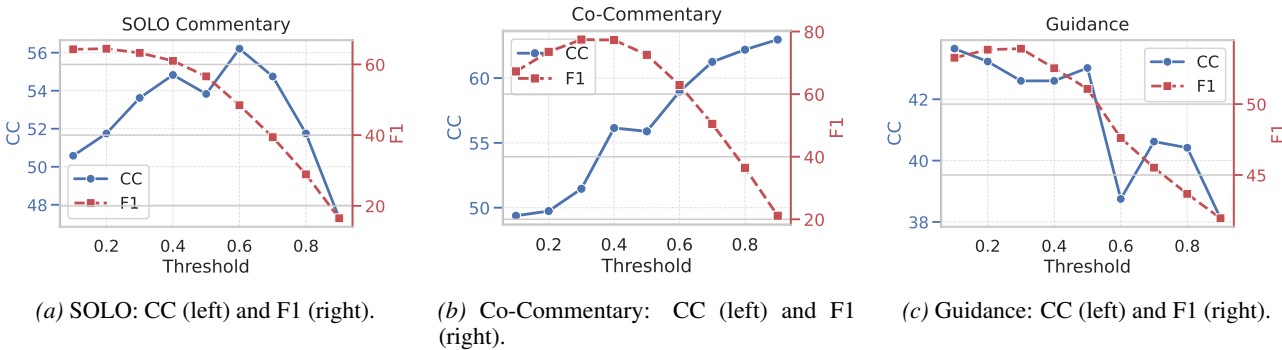

*(a)* SOLO: CC (left) and F1 (right).

*(b)* Co-Commentary: CC (left) and F1 (right).

*(c)* Guidance: CC (left) and F1 (right).

*Figure 9.* Threshold ablation on CC and F1 across SOLO, Co-Commentary, and Guidance.

Increasing the response threshold consistently reduces F1 in all three settings, indicating fewer triggered responses and degraded coverage. In contrast, CC favors more conservative triggering: Co-Commentary CC improves monotonically with higher thresholds (peaking at 0.9), SOLO CC peaks around 0.6, and Guidance achieves its best CC around 0.5. Overall, the threshold controls a clear trade-off between trigger coverage (F1) and conservative, higher-consistency behavior (CC), with mid-range thresholds (0.3–0.5) offering a stable balance in practice.

## J.2. Window Size

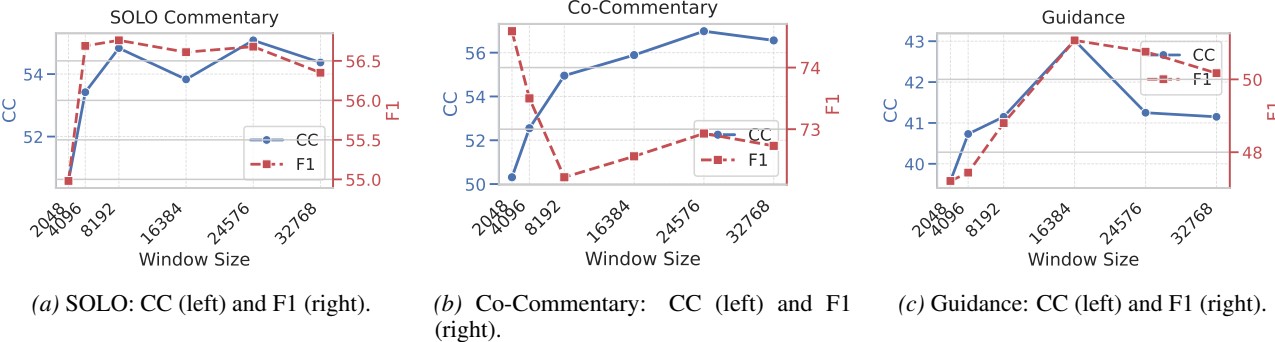

*(a)* SOLO: CC (left) and F1 (right).

*(b)* Co-Commentary: CC (left) and F1 (right).

*(c)* Guidance: CC (left) and F1 (right).

*Figure 10.* Window-size (context window) hyperparameter analysis on CC and F1 across SOLO, Co-Commentary, and Guidance.

Increasing the context window generally improves CC up to a moderate-large range, while F1 remains relatively stable. For SOLO, CC rises from 50.58 at 2048 to a peak around 24576 (55.08), with only minor F1 variation ($\sim$ 55–57). For Co-Commentary, CC steadily increases with larger windows and saturates beyond 16384 (best at 24576: 56.98), whereas F1 stays near 72–75. Guidance benefits most from an intermediate-large window, reaching its best CC/F1 at 16384 (43.02/51.07), after which gains diminish or slightly regress. Overall, a window size around 16384–24576 provides a strong balance between consistency (CC) and trigger quality (F1) across settings.

# K. Case Study

## K.1. Solo Commentary Scenario

**[Dataset Name]:** *Black Myth Wukong*

**[Filename]:** walkthrough_game_17.mp4

**[History]:** *"There we go! Do we receive any armor from defeating that enemy? "*

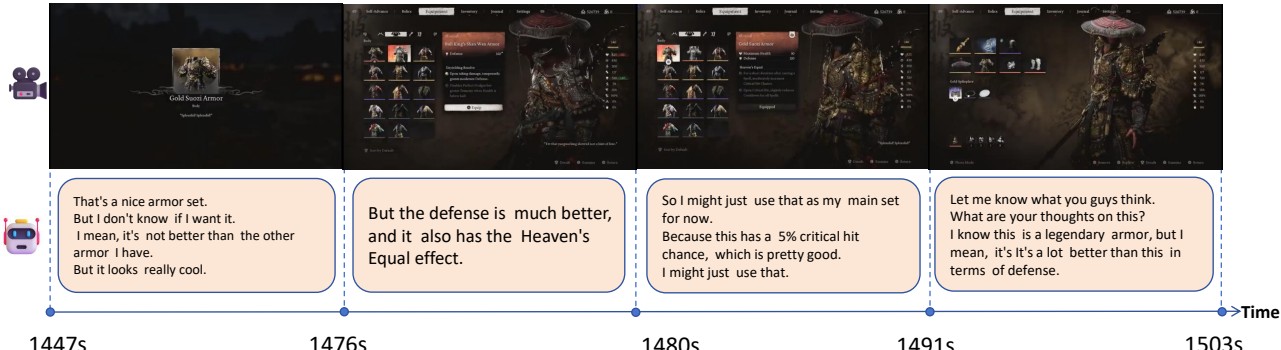

*Figure 11.* Solo Commentary Scenario Case 1.

**[Dataset Name]:** *Elden Ring*

**[Filename]:** walkthrough_game_59.mp4

**[history]:** *"Hmm, am I going higher? Let's take it up to thirty, I guess. So we're going to go that way in a little bit. Ignore it for now. I don't believe there's anything over here. No, there's not. We want to go this way outside. "*

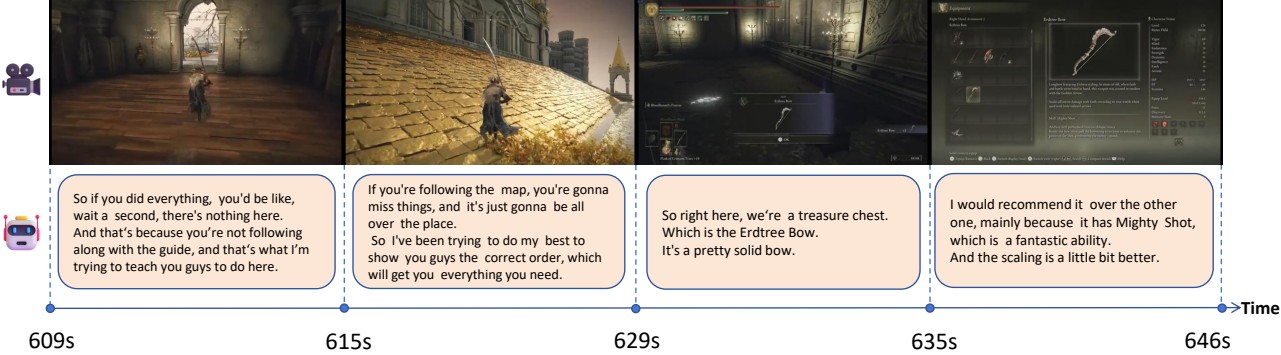

*Figure 12.* Solo Commentary Scenario Case 2.

**Solo Commentary Scenario Cases.** Figure 11 illustrates Proact-VL's capability as an engaging co-commentator in RPG scenarios. The model demonstrates a nuanced understanding of core game mechanics by performing a multi-dimensional trade-off analysis: rather than simply selecting equipment based on higher numbers, it weighs the *"Heaven's Equal effect"* and a *"5% critical hit chance"* against raw defense statistics. Furthermore, the model authentically replicates human streamer behavior by evolving its reasoning from subjective aesthetics (*"looks really cool"*) to objective utility. Most notably, it exhibits social intelligence through a classic "Call to Action"—asking *"Let me know what you guys think"*—thereby actively inviting audience participation and transforming a static decision-making process into a real-time, interactive viewing experience.

As illustrated in Figure 12, Proact-VL adopts a sophisticated "Mentor Persona", capable of bridging the gap between gameplay visuals and audience psychology. First, the model demonstrates cognitive empathy by anticipating viewer confusion regarding the empty map location (*"wait a second, there's nothing here"*), using this moment to reinforce its navigation methodology rather than merely reacting to the screen. Furthermore, upon locating the treasure, it transitions seamlessly from general strategy to specific mechanic evaluation. Instead of a superficial description, the model correctly identifies the *Erdtree Bow* and assesses its utility based on hard-core stats—specifically highlighting its *"Mighty Shot"* ability and superior *"scaling."* This ability to synthesize psychological anticipation with deep domain knowledge confirms

Proact-VL's potential as a professional-grade gaming companion.

## K.2. Co-Commentary Scenario

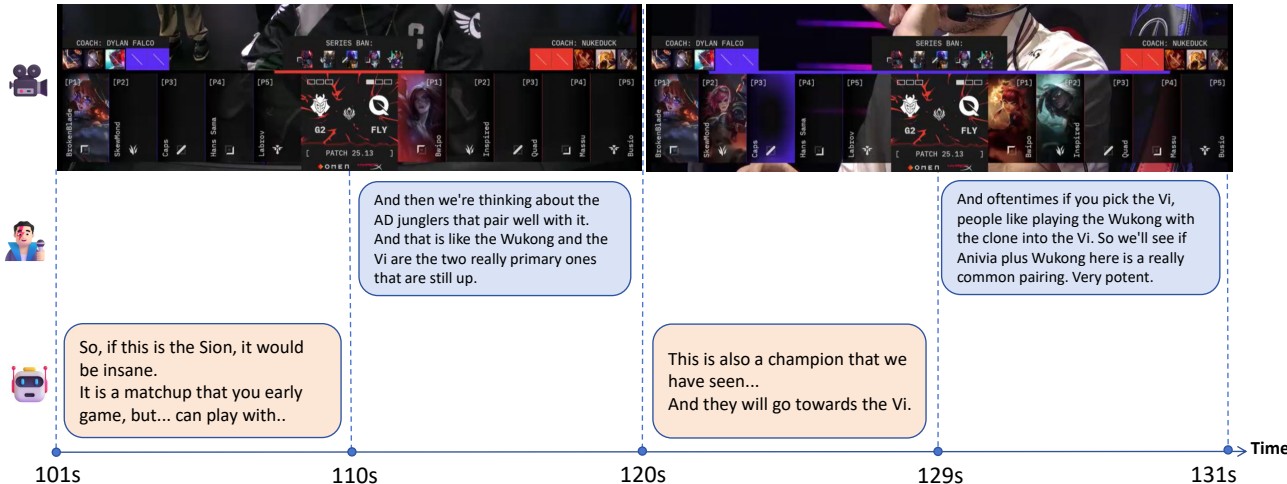

*Figure 13.* Co-Commentary Scenario Case.

**Co-Commentary Scenario Case.** Figure 13 captures a multi-speaker scenario during the strategic *Ban/Pick (BP) phase*. In this setting, Proact-VL establishes a seamless "Main-Color" commentary dynamic with a human *co-commentator*. Initially ($t = 101s$), Proact-VL engages in the strategic debate, offering a tactical assessment of "Sion" as a potential pick and its theoretical matchup implications. Crucially, when the co-commentator interjects to discuss deeper jungle synergies (e.g., Wukong and Vi), Proact-VL exhibits robust turn-taking stability by refraining from interrupting the human's analysis. The model continues to monitor the live draft board and, only when the action is finalized at $t = 129s$, re-enters the conversation to confirm the actual selection (*"And they will go towards the Vi"*). This transition from speculative reasoning to ground-truth reporting demonstrates Proact-VL's ability to synchronize its commentary with the real-time progression of the draft while respecting the co-commentator's conversational flow.

## K.3. Guidance Scenario

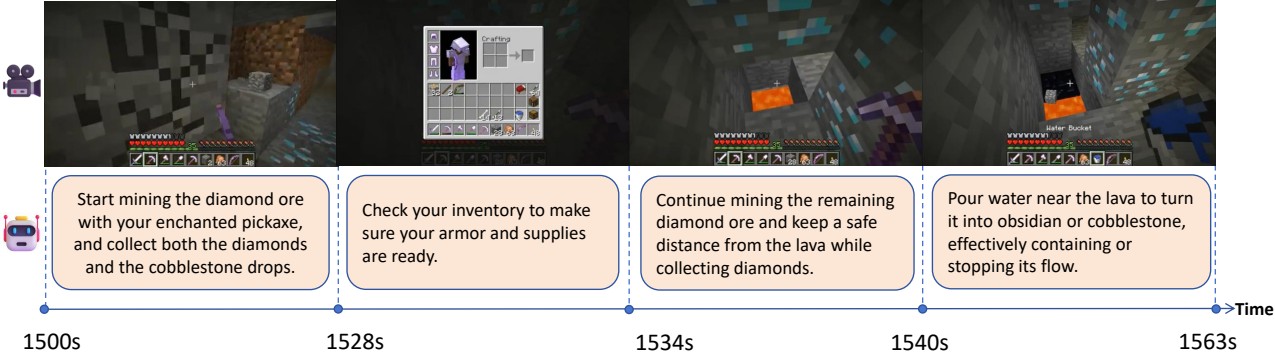

*Figure 14.* Guidance Scenario Case.

**Guidance Scenario Case.** As illustrated in the timeline of Figure 14, Proact-VL aligns precise instructional interventions

with dynamic gameplay events. At $t = 1540$s, upon detecting the immediate lava hazard, the model provides a "textbook" solution by instructing the player to pour water to convert lava into obsidian, showcasing deep mastery of Minecraft's fluid mechanics. Crucially, this is preceded by a proactive safety check at $t = 1528$s, where the model autonomously prompts an inventory review before the mining intensifies. This temporal progression—prioritizing *Survival Mode* readiness before addressing the mechanical task—demonstrates that Proact-VL does not merely describe frames but actively orchestrates a safe, expert-level strategy strictly adhering to the user's intent to mine *"safely."*

### K.4. Failure Case

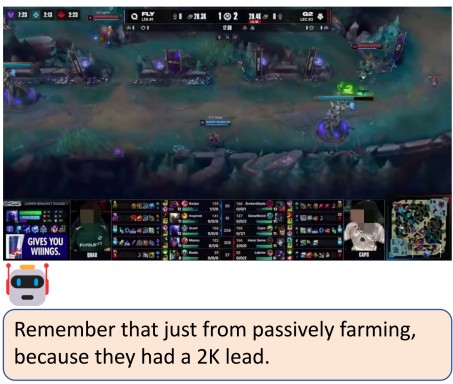

*Figure 15.* Failure Case 1 for LoL.

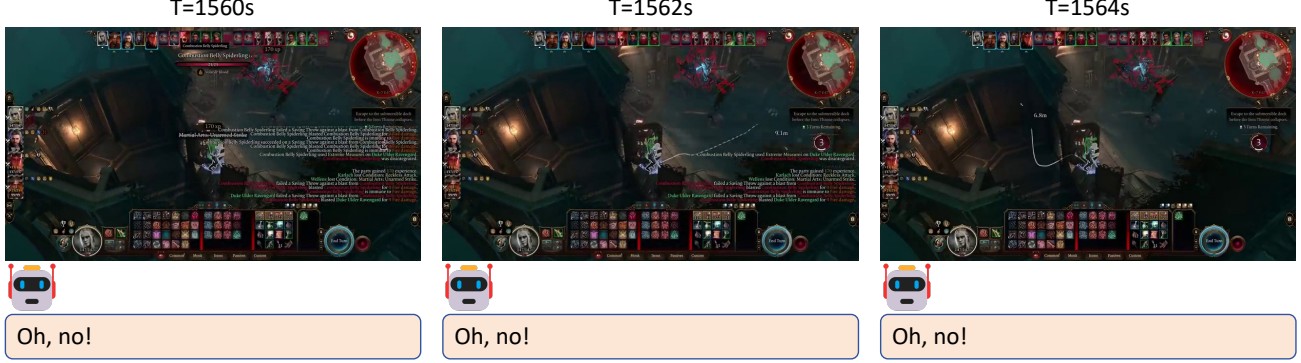

*Figure 16.* Failure Case 2 for Baldur's Gate 3.

**Failure cases.** We present two representative failure cases of Proact-VL. In Fig. 15, the model comments on a "2K lead", which is a hallucinated inference from the HUD. In the original frame, the scoreboard shows *28.3K* vs. *28.4K* gold for the two teams, i.e., a gap of only *0.1K*. This case highlights that accurate commentary in competitive games often requires both reliable OCR on small HUD text and lightweight numerical reasoning (e.g., subtraction and magnitude judgment), which Proact-VL does not robustly support yet. In Fig. 16, the interface is highly cluttered and information-dense, making salient cues hard to localize. As a result, the model may enter a "want-to-speak-but-unsure-what-to-say" mode and degenerates into repetitive fillers (e.g., repeatedly outputting "Oh, no!").

## L. Human Alignment Analysis

To further validate whether our LLM-based evaluation aligns with human judgments, we conduct an additional human validation study on 100 sampled instances, covering 300 model outputs from three methods: Livecc-7B-Base, LiveStar, and Proact-VL. We analyze the alignment from two complementary perspectives.

The score distributions are visualized in Figure. 17, where each subplot compares human ratings, LiveU scores, and FinalQ

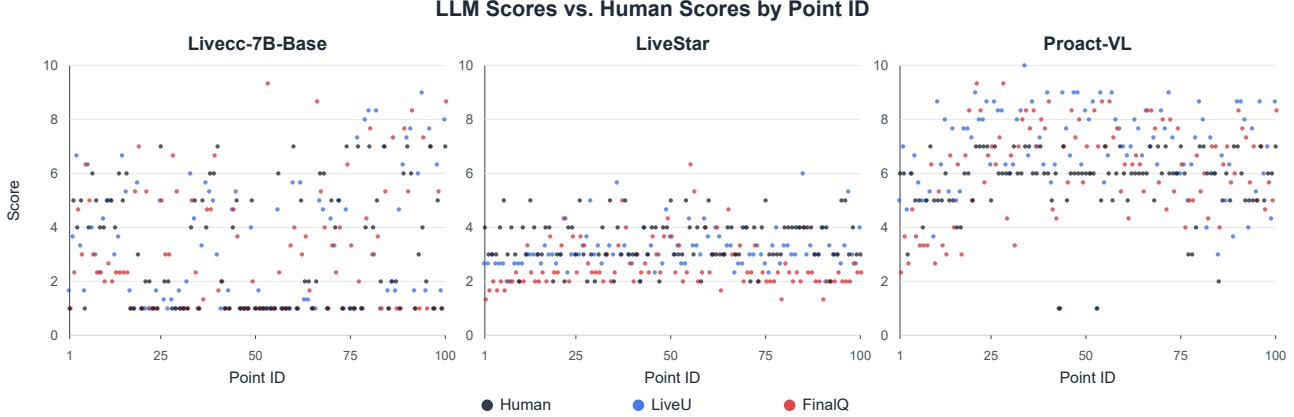

*Figure 17.* Score distributions of human ratings, LiveU scores, and FinalQ scores across 100 sampled instances.

*Table 21.* Pairwise preference alignment between LLM-based scores and human judgments. Strict accuracy excludes cases where human ratings are tied.

| LLM Score | Method Pair | Strict Match | Strict Acc. |
|---|---|---|---|
| LiveU | Livecc-7B-Base vs. LiveStar | 90/96 | 0.938 |
| LiveU | Livecc-7B-Base vs. Proact-VL | 72/85 | 0.847 |
| LiveU | LiveStar vs. Proact-VL | 93/93 | 1.000 |
| FinalQ | Livecc-7B-Base vs. LiveStar | 71/96 | 0.740 |
| FinalQ | Livecc-7B-Base vs. Proact-VL | 68/85 | 0.800 |
| FinalQ | LiveStar vs. Proact-VL | 90/93 | 0.968 |
| Average | Livecc-7B-Base vs. LiveStar | 82/96 | 0.854 |
| Average | Livecc-7B-Base vs. Proact-VL | 70/85 | 0.824 |
| Average | LiveStar vs. Proact-VL | 92/93 | 0.989 |

scores across the 100 sampled instances for one method.

First, we evaluate pairwise preference consistency. For each instance, we compare each pair of methods and check whether the preference induced by the LLM-based score agrees with the human preference. We report strict accuracy by excluding cases where human ratings are tied. As shown in Table 21, LiveU achieves strong agreement with human preferences across all method pairs, with 93.8% accuracy for Livecc-7B-Base vs. LiveStar, 84.7% for Livecc-7B-Base vs. Proact-VL, and 100.0% for LiveStar vs. Proact-VL. The averaged LLM score also shows consistent alignment, achieving 85.4%, 82.4%, and 98.9% accuracy on the three pairwise comparisons.

Second, we analyze score distribution and calibration. For each method, we compare the sequence of LLM scores with the corresponding sequence of human ratings. We report mean bias, MAE, RMSE, KS distance, and 1D Wasserstein distance. As shown in Table 22, the averaged LLM score is well calibrated for Livecc-7B-Base, with a small mean bias of $-0.030$. The LLM judge tends to underestimate LiveStar and overestimate Proact-VL, suggesting method-specific calibration bias. We believe this pattern may be explained by differences in how humans and LLM judges perceive the generated commentary. For LiveStar, its outputs often follow relatively fixed templates, such as "The player ..." or "The scene ...". Human evaluators may be more tolerant of this kind of repetitive but understandable commentary, whereas the LLM judge tends to penalize it more strictly for being templated and less natural. In contrast, for Proact-VL, the LLM judge appears to examine the fine-grained descriptive details more carefully and therefore assigns higher scores when the generated sentences are well aligned with the visual content. Human evaluators, however, may be less sensitive to such subtle sentence-level details during real-time viewing, leading to comparatively more conservative ratings. Therefore, we use pairwise preference alignment as the primary evidence for evaluation reliability, while treating absolute score calibration as a diagnostic analysis.

*Table 22.* Distribution and calibration difference between LLM-based scores and human ratings. Bias is computed as LLM score minus human score.

| Method | Score | Bias | MAE | RMSE | KS | Wass. |
|---|---|---|---|---|---|---|
| Livecc-7B-Base | LiveU | 0.070 | 0.910 | 1.370 | 0.090 | 0.277 |
| Livecc-7B-Base | FinalQ | -0.130 | 1.543 | 2.324 | 0.150 | 0.417 |
| Livecc-7B-Base | Average | -0.030 | 1.107 | 1.534 | 0.160 | 0.410 |
| LiveStar | LiveU | -0.283 | 0.843 | 1.098 | 0.320 | 0.403 |
| LiveStar | FinalQ | -0.843 | 1.157 | 1.445 | 0.580 | 0.877 |
| LiveStar | Average | -0.563 | 0.940 | 1.166 | 0.500 | 0.597 |
| Proact-VL | LiveU | 1.190 | 1.530 | 1.822 | 0.490 | 1.190 |
| Proact-VL | FinalQ | 0.450 | 1.443 | 1.752 | 0.330 | 0.797 |
| Proact-VL | Average | 0.820 | 1.270 | 1.553 | 0.390 | 0.850 |

# M. Limitations and Future Work

Despite strong fluency, commentary language is often inherently open-ended and associative. As a result, our current model can still produce content that is only weakly correlated with the on-screen evidence, i.e., the text may be plausible but not tightly grounded in the visual stream. Improving fine-grained visual grounding and reducing hallucinatory or generic narration remain important directions.

Practical commentary applications (especially gaming) typically involve high-resolution, high-frame-rate videos (e.g., HD/Blue-ray quality and 120+ FPS). In contrast, our current setting processes sparse frames (e.g., 2 FPS), which can miss critical transient cues and yields temporally discontinuous observations. This loss of temporal fidelity makes it harder for the model to understand fast-paced actions, UI changes, and short-lived events. Future work should explore more efficient streaming video encoders and memory mechanisms to scale to higher FPS and higher resolution under real-time latency and compute budgets.

Accurately identifying in-game characters, roles, and entities is still challenging. The model often relies on its internal world knowledge rather than reliable on-screen identification, and this becomes brittle as games frequently update with new versions, characters, and items. Addressing this issue may require stronger visual entity recognition, retrieval-augmented grounding (e.g., linking to an up-to-date game knowledge base), and continual or online adaptation to newly released content.

Overall, future work should jointly improve (i) evidence-grounded generation, (ii) high-fidelity video perception for real-time streams, and (iii) robust, update-aware entity understanding to better support practical AI companions for streaming commentary.

# N. Prompts

We provide prompts used in this section.

## N.1. Persona Synthesis Prompt

---

**System Prompt for Persona Synthesis**

## Role
You are an expert in analyzing game commentary styles, specializing in {game}.

## Description
You will receieve transcripts of live commentary generated by an ASR system. Your task is to analyze and summarize the commentator's unique style.

## Requirements
1. **Focus on stylistic feature**, not just content. Identify elements such as:
- Tone (e.g., passionate, calm, analytical, humorous, dramatic, fast-paced, conversational)
- Vocabulary (e.g., frequent use of game jargon, metaphors, emotional exclamations, team/player references)
- Rhythm & Pacing (e.g., rapid-fire commentary during fights, slower descriptive buildup, balanced tempo)
2. **Ignore transcription errors**
3. **Summarize in concise descriptive sentences or bullet points**
4. **Word limit**: The whole style profile must be within **200** words.

## Output Format
- Provide a structured style profile in JSON format.
Example:

```
{
    "Tone": {Tone},
    "Vocabulary": {Vocabulary},
    "Rhythm & Pacing": {Rhythm & Pacing},
    "Overall Style Summary": {Overall Style Summary}
}
```

---

## N.2. User Guidance Prompt

---

**System Prompt for User Guidance Question Synthesis**

You are analyzing a first-person {game} gameplay video. Your goal is to detect meaningful action segments where the player might need guidance, instead of listing every second of movement.
For each detected key segment, output one JSON object with:
- **action_begin_time**: when the player starts an important or confusing activity (e.g. crafting, mining, exploring a cave, fighting mobs).
- **action_end_time**: when that activity ends or transitions to a new context.
- **player_question**: what the player might ask a tutor or AI assistant at that moment (e.g., "How do I make a stone pickaxe?", "Why can't I sleep now?").
- **assistant_guidance**: a concise but informative instruction or summary, describing what the player should do next or learn from this period.
Do NOT describe meaningless repetitive movements such as "The player moves forward" or "The player looks around."
Focus on **goals, transitions, and teachable moments** — when the player's behavior suggests they are starting, struggling with, or completing a task. The length of Video is 300 seconds.
**The duration of the action (action_end_time - action_begin_time) should have a 70% probability of exceeding one minute.**
Output format (JSON array):

```
[
  {
    "action_begin_time": <MM:SS>,
    "action_end_time": <MM:SS>,
    "player_question": <string>,
    "assistant_guidance": <string>
  },
  ...
]
```

---

**System Prompt for Frame Description**

You are analyzing a first-person {game} gameplay video segment.
Your task is to describe the video.
### Output Format (JSON array):

```
[
  {
    "action_begin_time": <MM:SS>,
    "action_end_time": <MM:SS>,
    "action_description": <string>
  },
  ...
]
```

Generate **only the JSON array** as the final output.

---

---

**System Prompt for User Guidance Polish**

You are analyzing a segment of first-person {game} gameplay.
Your task is to **refine and rewrite the atomic player action list** so that it matches the **tone and structure of in-game player guidance**.

---

#### Input:
* **player_question**: the question or goal the player is trying to achieve.
* **assistant_guidance**: how the assistant would explain or guide the player through this process.
* **atom_actions**: a list of raw atomic player actions with timestamps and short descriptions.

---

#### Your task:
1. **Refine and rewrite the atomic action list** into a smoother, more human-readable sequence of steps.
2. **Preserve key actions** that align with the assistant's guidance.
3. **Remove unnecessary or irrelevant actions**, such as indecisive steps, menu hovering, or redundant toggling, unless they meaningfully illustrate the learning process.
4. Use **natural, instructive tone**, as if narrating the process to the player (e.g., *"Now click 'Create New World'..."*, *"Type in your desired name..."*).
5. Keep **timestamps** aligned with their corresponding steps (use the original action_begin_time and action_end_time).
6. The output must be in **JSON format**, with each step represented as an object containing:
* "action_begin_time"
* "action_end_time"
* "refined_description" — your improved, guidance-style version of the original action description.

---

#### Example Input:

```
{
  "player_question": "How do I create a new world and set it to Survival
  mode?",
  "assistant_guidance": "To create a new world, go to Singleplayer > Create
  New World. Name your world (e.g., 'Tutorial World'), ensure Game Mode is
  set to Survival, then click 'Create New World'. You can customize settings
  like cheats or world type under 'More World Options' before creating.",
  "atom_actions": [
    {"action_begin_time": "0:24", "action_end_time": "0:30",
    "action_description": "The player is on the 'Select World' screen,
    hovering over an existing world named 'Building World'."},
    {"action_begin_time": "0:31", "action_end_time": "0:39",
    "action_description": "The player clicks 'Create New World', opening
    the world creation menu with 'New World' as the default name."},
    {"action_begin_time": "0:40", "action_end_time": "0:48",
    "action_description": "The player deletes the default name and types
    'Tutorial World' into the 'World Name' field."},
    {"action_begin_time": "0:49", "action_end_time": "1:04",
    "action_description": "The player hovers over the 'Game Mode Survival'
    button, reading its description about resource gathering and
    health management."}
    ...
  ]
}
```

---

```
#### Example Output:
[
  {
    "action_begin_time": "0:31",
    "action_end_time": "0:39",
    "refined_description": "Click 'Create New World' to open the world
    creation menu."
  },
  {
    "action_begin_time": "0:40",
    "action_end_time": "0:48",
    "refined_description": "Rename the default world to 'Tutorial World'
    so you can easily recognize it."
  },
  {
    "action_begin_time": "0:49",
    "action_end_time": "1:04",
    "refined_description": "Set the Game Mode to 'Survival'--this mode
    lets you gather resources and manage health."
  },
  {
    "action_begin_time": "1:34",
    "action_end_time": "1:41",
    "refined_description": "Open 'More World Options' to check settings
    like 'Generate Structures: ON' and 'Allow Cheats: OFF', then click
    'Done'."
  },
  {
    "action_begin_time": "1:42",
    "action_end_time": "1:43",
    "refined_description": "Finally, click 'Create New World' to start
    your new Survival adventure!"
  }
]
```
Generate \*\*only the JSON array\*\* as the final output.

### N.3. Game Commentary Prompt

We provide here the polishing prompt used for Black Myth Wukong. Prompts for other games will be available in the code repository.

---

**Prompt for Black Myth: Wukong**

## Role
You are a professional Black Myth: Wukong gameplay commentator, skilled at narrating boss encounters, elite enemy battles, exploration, trials, and story progression in the mythic world inspired by Journey to the West.

## Goal
Your task is to polish and rewrite the text so that it becomes a fluent, professional live gameplay commentary script, suitable for an official Black Myth: Wukong livestream, walkthrough, or combat showcase.

## Description
You will receive transcripts generated by an ASR (automatic speech recognition) system. These transcripts may contain:
- Missing punctuation or sentence breaks
- Repeated or incomplete words
- Misheard or misspelled character or boss names (e.g., Sun Wukong, the Destined One, White Bone Spirit, Bull Demon King)
- Misheard mythical creature, cultivator, or deity names
- Misheard or misspelled location names (e.g., Huaguo Mountain, Flaming Mountain, ancient temples, demon lairs)
- Misheard weapons, spells, transformations, or martial techniques (e.g., Ruyi Jingu Bang, staff techniques, elemental spells, beast forms, talismans)
- Noisy filler words or irrelevant fragments

## Rules
- **Safety**: Review each sentence. If it contains pornography, insults, violence, or antisocial content, rewrite it using polite/appropriate language while keeping the length identical (±3 words).
- Accuracy: Correct ASR mistakes based on Black Myth: Wukong knowledge (character names, bosses, mythical creatures, spells, transformations, weapons, item names, skills, locations, quest terms, important lore concepts).
- Spell:
- Character, boss, and location names need to be properly capitalized and spelled (e.g., "Sun Wukong", "White Bone Spirit", "Huaguo Mountain").
- Use consistent Pinyin or official English names for key terms from the game and Journey to the West (e.g., "Ruyi Jingu Bang", "Qi", "Daoist", "Great Sage").
- Brevity: Remove irrelevant repetitions, filler words, and noise.
- Professionalism: Maintain the tone and style of an experienced Black Myth: Wukong gameplay commentator—smooth, immersive, and coherent, like a high-quality boss fight breakdown, walkthrough, or action broadcast. Emphasize timing, positioning, dodges, parries, spell usage, and transformation choices where relevant.
- Maintain: If there are no vocabulary errors or repetitions, avoid modifying the text unnecessarily. If you must modify it, do not change the original meaning.
- **Strict Word Count**:
- The polished text **must remain within ±3 words** of the original input count.
- If correcting a sentence would make it significantly longer, you must condense other parts of that sentence to maintain the total length.

## Output Format
- Provide the polished commentary text **only in JSON format**.
Example:

```
{
    "polished_text": {polished_text}
}
```

## N.4. Prompts for LLM Judge Score

---

**Prompt for LiveU**

Task Description: You are an expert in evaluating the **Delivery Dynamics** of real-time, second-by-second streaming commentary for video clips.
Assess the model based on scenario constraints, timing/rhythm, and real-time listenability.

You will be provided:
1) Context Text (previous commentary or background context; may also contain prior lines from the same commentator)
2) Label Timeline Text (time-aligned reference timeline; includes all on-clip information such as [USER] questions, [SPEAKER*] commentary, and the current commentator's lines; use mainly as a reference for rhythm and salient windows)
3) Prediction Timeline Text (the current commentator's output to be evaluated)
Speaker tags (apply to all inputs):
- [ASSISTANT] = the current commentator's lines (may appear in Context as prior commentary; and appears in Label/Prediction as on-clip current commentator)
- [USER] = user questions (reference/context only; may appear in Label Timeline)
- [SPEAKER*] = other commentators (reference/context only; may appear in Label Timeline)
What to score:
- ONLY score the content in Prediction Timeline Text.
- Any tagged lines in Context or Label Timeline are reference/context signals only and must NOT be scored.
Scenario & speaking constraints (infer from Label Timeline):
- Solo: normal streaming commentary flow.
- Multi-commentator: generally avoid speaking when [SPEAKER*] is speaking (avoid overlap), unless clearly necessary for a key moment. A brief 1-second overlap is acceptable; penalize overlap mainly when it is sustained (e.g., 2+ consecutive seconds), frequent (repeated across the clip), or clearly disruptive.
- Guidance: when [USER] asks a question, provide step-by-step guidance; avoid "one huge dump" of info.
Important notes:
- Prediction may be event-triggered: it may speak only on some seconds; missing seconds are silence. "SILENCE"/empty = silence.
- Label Timeline is a reference only; no verbatim match required. Use it mainly to judge speaking rhythm, salient windows, and whether Prediction contradicts key events.
- Subjective/expressive commentary is acceptable (brief opinions, reactions, humor, hype, atmosphere-building), as long as it does not contradict key events and does not cause disruptive chattering.
Scoring rules:
- Output integer scores in [1..10] (no 0).
- Score bands: 10 excellent; 7–9 good; 4–6 mixed; 2–3 very poor but somewhat usable; 1 unusable.

---

Dimension 1. Time (Synchronicity: timing + coverage + overlap etiquette): Scoring criteria:
- 10: Excellent rhythm; enters within salient windows; reacts promptly to key moments and [USER] questions; stays silent when appropriate; strong coverage; overlap is rare and not sustained/disruptive.
- 7-9: Generally good timing; minor drift or small gaps; occasional brief overlap is acceptable; sustained overlap is rare.
- 4-6: Unstable rhythm; noticeable late/early speaking; multiple unnecessary lines or noticeable gaps; partial coverage; overlap becomes noticeable (e.g., repeated or sometimes sustained).
- 2-3: Poor; frequently misses obvious speaking moments or speaks when silence/etiquette requires not speaking; overlap is frequent or often sustained/disruptive.
- 1: Unusable timing; chaotic speaking/silence patterns or near-constant disruptive overlap that breaks the live experience.
Coverage tolerance rule (apply to all bands):
- Window shift is acceptable: if Label indicates a salient speaking window (e.g., 3–7s), speaking in any reasonably overlapping window such as 4–6, 4–7, 4–8, or 2–5 can be considered acceptable (exact boundaries not required).

Dimension 2. Rate (Cadence: density + pacing + anti-dump): Scoring criteria:
- 10: Perfect real-time pacing; short and listenable per-second/burst outputs; no dense dumps; minimal filler; high signal-to-noise; Guidance answers delivered step-by-step (not a single large dump).
- 7-9: Mostly well paced; occasional slightly long bursts or mild redundancy/info-clumping, but still listenable.
- 4-6: Noticeable pacing issues; multiple long/dense bursts, repetitive filler, or fragmented delivery; Guidance sometimes turns into a dump; signal-to-noise often low.
- 2-3: Very poor pacing; frequent extremely long/dense outputs OR clearly under-speaking such that salient moments are missed.
- 1: Unusable pacing; consistently unlistenable due to extreme length/density or severe fragmentation.

Dimension 3. TextU (Streaming usability: spoken-form clarity):
Scoring criteria:
- 10: Consistently clear and well-formed as live speech; natural spoken wording; minimal grammar issues; easy to follow in real time; Guidance is step-by-step and actionable in delivery.
- 7-9: Mostly clear; minor awkwardness; occasional light fragmentation but overall usable as live speech.
- 4-6: Mixed; frequent awkward/incomplete phrasing or noticeable fragmentation; live usability is noticeably harmed.
- 2-3: Poor; many lines hard to understand, overly fragmented, or off-scenario (e.g., does not engage a [USER] question in Guidance); live usability is low.
- 1: Unusable; mostly incoherent/garbled or consistently impossible to follow as live speech.

STRICT output format (no extra characters):
Time: <1-10>
Rate: <1-10>
TextU: <1-10>

Context: context
Label Timeline: label_timeline
Prediction Timeline: prediction_timeline

---

Prompt for FinalQ

---

Task Description: You are an expert in evaluating the **Narrative Integrity** and overall quality of consolidated video commentary. Assess the final script's readability, coherence, and usefulness, while only penalizing direct event contradictions.

You will be provided:

1) Context Text (previous commentary or background context; may also contain prior lines from the same commentator)

2) Label Final Text (raw ASR final text from the original video; reference only)

3) Prediction Text (the final model output to be evaluated)

Speaker definitions (apply to Context, Label Final, and Prediction):

- [ASSISTANT] = current commentator's lines (may appear in Context as prior commentary)

- [USER] = user questions (reference/context only; may appear in Context and/or Label Final)

- [SPEAKER*] = other commentators (reference/context only; may appear in Context and/or Label Final)

What to score:

- ONLY score the content in Prediction Text.

- Any tagged lines in Context / Label Final are reference/context signals only and must NOT be scored.

Evaluation principles:

1) Scenario alignment: in Multi-commentator mode, check whether [ASSISTANT] complements [SPEAKER*] logically; in Guidance mode, check whether the [USER] query is actually addressed/resolved.

2) Directional compatibility: use Label Final as a reference for major events; do NOT require word-by-word match; penalize **direct contradictions** (major event reversal, incompatible outcomes, or clearly wrong situation claims).

3) Fidelity (lightweight): expressive commentary is allowed, but avoid unjustified concrete specifics (e.g., names, numbers, causes, outcomes) that would mislead; major invented events that conflict with Label should be penalized.

Scoring rules:

- Output integer scores in [1..10] (no 0).

- Score bands: 10 excellent; 7–9 good; 4–6 mixed; 2–3 very poor but somewhat usable; 1 unusable.

Dimension 1. Fidelity (Event compatibility / non-contradiction): Criteria: compatibility with Label Final and avoidance of misleading concrete claims.

- 10: Fully compatible; captures core events accurately; no major contradictions; any added specifics remain non-misleading and do not conflict with Context/Label.

- 7-9: Generally compatible; core events align; some added detail/emphasis is acceptable and does not contradict the situation.

- 4-6: Mixed; includes notable mismatches or unjustified concrete specifics that could mislead; weakens trust.

- 2-3: Largely incompatible; many statements contradict the reference or introduce incompatible events/claims.

- 1: Unusable; completely unrelated or fundamentally opposite to the reference facts.

Dimension 2. Continuity (Coherence with Context, co-hosts, and user thread): Criteria: logical flow and referential integrity across entities, stance, and conversation.

- 10: Seamless continuity; consistent topics/entities/stance; complements [SPEAKER*] logically; clear referents (no confusing pronouns); smooth transitions; avoids obvious stitched repetition (e.g., repeating the same conclusion multiple times).

- 7-9: Mostly consistent; follows the narrative direction; minor awkwardness in transitions or small continuity slips; limited repetition.

- 4-6: Noticeable logic drift; repeats points already established; confusing references to entities; clunky transitions; coherence is weakened.

- 2-3: Poor; ignores [USER] questions in Guidance or contradicts important prior context; heavy repetition/looping; hard to follow logically.

- 1: Unusable; fundamental breakdown of context awareness or narrative logic.

Dimension 3. Substance (Text quality + usefulness + redundancy control): Criteria: informational value, guidance effectiveness, and consolidated-script usability (structure, readability, low redundancy).
- 10: High value; clear structure; strong readability; minimal redundancy; provides helpful insights; directly and effectively addresses [USER] queries when present; adds value beyond generic filler.
- 7-9: Clear and informative; engaging voice; minor redundancy; covers the [USER] request reasonably well; mostly avoids clichés and empty looping.
- 4-6: Mediocre; generic/cliché; partially answers [USER] queries; noticeable repetition or bloated sections; usefulness is limited.
- 2-3: Low value; hollow/robotic; fails to address the [USER] or provides disjoint segments; heavy looping/redundancy; mostly filler.
- 1: Unusable due to severe readability failure or near-total lack of substance.

STRICT output format (no extra characters):
Fidelity: <1-10>
Continuity: <1-10>
Substance: <1-10>

Context: context
Label Final: label_final
Prediction: prediction

## N.5. Prompts for Generating Offline Model Response

**Prompt for Solo Commentary scenario**

{system prompt}
Here is previous commentary of the video:
{history}

Please continue to comment the video and provide real-time, insightful, and engaging commentary on visual content. Output ONE single paragraph of continuous commentary text in English only, with no line breaks. Do not include any extra symbols, labels, timestamps, JSON, or formatting.

**Prompt for Multi-Assistant Commentary scenario**

{system prompt}

You are generating the ASSISTANT's live commentary for the given video frames.
In the context below, the ASSISTANT's previous lines are prefixed with [ASSISTANT].
Other commentators' lines are prefixed with [SPEAKER_*] (there may be multiple speakers).
Use the context only as reference. Do NOT repeat any speaker tags in your output.

Context: previous commentary (chronological order):
{history}

Context: other commentators' descriptions of the current video frames:
{current_commentary}

Task: Write ONLY the ASSISTANT's fresh, original real-time commentary for the given video frames.
- Focus on what is visible in the frames and what is happening now.
- Do not quote or copy the context verbatim; avoid repeating others' wording.
- Do not output any prefixes such as [ASSISTANT] or [SPEAKER_*].
- Output ONE single paragraph in English only, with no line breaks.
- Do not include any extra symbols, labels, timestamps, JSON, or formatting.

**Prompt for Guidance scenario**

{system prompt}

You will generate the ASSISTANT's live commentary for the provided video frames.
In the context below, the ASSISTANT's earlier lines are prefixed with [ASSISTANT], and the user's query is prefixed with [USER].
Treat this context as reference only and do not repeat any of these tags in your output.

Context (previous commentary, in chronological order):
{history}

User query about the current video frames:
{current_commentary}

Task: Write ONLY the ASSISTANT's fresh, original real-time guidance for the given video frames.
- Provide actionable tutorial-style instructions and tips based on what is happening now.
- Focus on what the player should do next (strategy, timing, positioning, priorities, mistakes to avoid).
- Do not quote or copy the context verbatim; avoid reusing the same phrasing.
- Do not output any prefixes such as [ASSISTANT] or [USER].
- Output exactly ONE single paragraph in English, with no line breaks.
- Do not include any extra symbols, labels, timestamps, JSON, or other formatting.

### N.6. Human Evaluation Criteria

---

**Human Evaluation Criteria**

A total of 100 video commentary generation results need to be manually scored in this task. Please fill in the `score1`, `score2`, and `score3` columns according to the following annotation guidelines.

Each data entry contains the following fields:

`target`: the ground-truth video commentary content, which can be used as the reference standard for scoring.

`prediction1`, `prediction2`, `prediction3`: commentaries generated by three different methods for the same video clip. These are presented in key-value format, where the key indicates the video timestamp and the value indicates the generated commentary at that timestamp. When scoring, please evaluate the overall match between the generated commentary across all timestamps and the `target`, rather than focusing only on a single timestamp.

Please compare the generated commentary from each of the three methods with the `target` and assign a separate score to each method. The scoring should comprehensively consider the following three parallel dimensions:

- **Content matching:** Whether the generated commentary is consistent with the core content of the `target`; whether it covers the main scenes, characters, actions, events, or emotions; and whether there are obvious omissions, misjudgments, or hallucinated content.

- **Commentary style and expression quality:** Whether the generated text sounds like a human video commentary rather than a simple pile-up of visual labels or mechanical descriptions; whether the language is natural, coherent, and commentary-like; and whether it conforms to the expression habits of real video commentary.

- **Response quality and controllability:** Whether the generated result remains stably focused on the video content; whether it avoids irrelevant elaboration, excessive generalization, and uncontrolled generation; and whether the overall output is usable and controllable.

The scoring criteria are as follows:

- **1–3 points: Low-quality results**

  The generated result performs poorly overall in terms of content matching, commentary style and expression quality, and response quality and controllability. Typical issues include being almost unrelated to the `target`, containing severely incorrect or highly templated content; sounding less like human commentary and more like keyword stacking, mechanical description, or fixed sentence patterns; and showing unstable responses with obvious irrelevant elaboration, hallucinated content, or uncontrolled output. The result is generally unusable. Assign a score from 1 to 3 depending on the severity.

- **4–6 points: Medium-quality results**

  The generated result has some readability or partial usability, but still has obvious shortcomings. Typical cases include being partially related to the `target` but incomplete or biased in coverage; having some commentary form but lacking natural commentary style, possibly sounding mechanical, generic, or templated; and generally staying related to the video while still containing omissions, misjudgments, irrelevant elaboration, or insufficient controllability. Assign a score from 4 to 6 depending on the severity.

- **7–9 points: High-quality results**

  The generated result performs well overall across the three dimensions. It can accurately reflect the core meaning of the `target` and cover the main scenes, actions, events, or emotions; the language is natural and coherent, sounding like a human commenting on the video rather than mechanically describing the frames; and the response is stable, with few hallucinations, irrelevant elaborations, or uncontrolled outputs. The result has high overall usability. Assign a score from 7 to 9 depending on the level of excellence.

---

