# OpenReview forum: "Proact-VL: A Proactive VideoLLM for Real-Time AI Companions"
_ICML.cc/2026/Conference — ICML 2026 regular_

### Official Review · Reviewer_Avhb · 2026-03-02

**Soundness:** 3
**Presentation:** 3
**Significance:** 3
**Originality:** 3
**Overall Recommendation:** 4
**Confidence:** 3

**Summary:**

This paper aims to achieve low-latency, proactive, and human-like interaction in real-time streaming video scenarios. The authors present Proact-VL, a framework that introduces a chunk-wise input-output schema, coupled with a lightweight proactive mechanism that autonomously decides when to speak based on a "FLAG" token's hidden state. The authors also curate the Live Gaming Dataset, a 561-hour collection spanning 12 diverse game titles and three interaction modes: solo commentary, co-commentary, and user guidance. Extensive experiments on their new Live Gaming Benchmark demonstrate that Proact-VL significantly outperforms existing models like VideoLLM-online and LiveCC in both response timing accuracy (TimeDiff) and overall text quality

**Compliance With Llm Reviewing Policy:**

Affirmed.

**Key Questions For Authors:**

see weakness

**Limitations:**

see weakness

**Strengths And Weaknesses:**

**Strength:**

1.	The paper introduces a unique proactive "FLAG" token mechanism. Unlike prior models that either generate long responses once triggered or speak at fixed intervals, Proact-VL learns to decide per-second whether to speak or remain silent, enabling more natural, human-like pacing.

2.	The methodology is technically robust, addressing practical streaming issues such as infinite inference through selective KV cache eviction and a "reverse-RoPE" correction to maintain positional coherence.

3.	The Live Gaming Benchmark could be a  standardized evaluation framework for proactive agents

**Weakness:**

1.	The data processing procedure is heavily relied on MLLMs. However, quality control in each stage is not shown in the current version, raising concern about data annotation quality and possible model bias.

2.	The performance of the proactive mechanism is highly sensitive to the trigger threshold ($\tau$). As noted in the case study, low thresholds lead to severe over-triggering, while high thresholds result in total silence, suggesting that the model's "innate" sense of timing still requires significant manual calibration for different scenarios.

3.	The paper claims to achieve superior response latency, but there is no latency comparison with similar methods.

---

> ### Author Rebuttal · Authors · 2026-03-31
>
> We thank the reviewer for the positive assessment and constructive suggestions. Below we respond to the three concerns on data quality, threshold sensitivity, and latency.
>
> > Weakness 1: Data Quality and Annotation Reliability
>
> We agree that data quality is important. We would like to clarify that MLLMs in our pipeline are used not only for data construction, but also for data cleaning and quality control, especially in the polish stage of the data pipeline shown in Figure 3. This step is part of our actual dataset construction process rather than something introduced only for rebuttal.
>
> Importantly, the **majority of our supervision comes from ASR-derived commentary**, which is grounded in real human commentary from source videos. This substantially alleviates the concern about possible model bias, because the main labels are based on one-to-one transcription/alignment of human speech rather than unconstrained model generation in open-ended settings.
>
> In practice, for ASR-derived commentary, we mainly observed two issues: WhisperX may make mistakes on game-specific terms such as character names, item names, and other domain-specific expressions; and raw commentary may contain profanity, slang, or other low-quality expressions. We address these issues through terminology correction, sensitive-word matching, rule-based normalization, synonym replacement, and LLM-based polishing/review.
>
> For tutorial-style QA, MLLMs are not only used to generate the initial annotation, but also used in the **polish stage**, as shown in Figure 3, to remove irrelevant actions, refine the text, and improve consistency with the source video. This is another way in which MLLMs contribute to data quality control rather than merely introducing annotation noise.
>
> For both ASR-derived commentary and tutorial-style QA, we apply a simple but effective quality-control pipeline: manual spot-checking against the original video, LLM-based review for safety, consistency, and text quality, automatic flagging of suspicious samples, and final human review of all flagged cases. In our current pipeline, the first-pass suspicious rate identified by the LLM reviewer is below 0.5%, and all flagged samples are then manually corrected. We will make this pipeline more explicit in the revision, as it is not sufficiently emphasized in the current version.
>
> > Weakness 2: Trigger Threshold Sensitivity
>
> We agree that the trigger threshold controls an important trade-off: lower thresholds make the model more talkative, while higher thresholds make it more conservative. We view this as a controllable operating point rather than a sign that the method only works under narrowly tuned settings.
>
> The low-threshold and high-threshold examples in Figure 5 are intentionally chosen as extreme cases. They do not represent the normal operating regime. As shown in Appendix I.1, performance remains relatively stable in a moderate threshold range (approximately 0.3–0.7), and clear degradation mainly appears near the extremes. This suggests that the method does not require heavy manual calibration in practice; a reasonable threshold range already works stably, while small adjustments allow different speaking styles. A dynamic adjustment strategy could better serve users by adapting the response frequency to different user preferences and interaction scenarios.
>
> > Weakness 3: Response Latency Comparison
>
> We thank the reviewer for pointing this out. In the revision, we add a direct latency comparison under the same hardware setting. The main conclusion is that Proact-VL has latency comparable to streaming real-time baselines such as LiveCC, while being clearly faster than the proactive/event-triggered baseline LiveStar.
>
> Specifically, we measure response latency on the LiveGamingBenchmark and report the results below. We use three practical statistics: response rate (RR), average token length under triggered responses (ATL), and average response time under triggered responses (ART). Here, ART measures the full response pipeline, including video reading, model processing, and generation of the response for the current second.
>
> | Model  | RR (%) | ATL | ART (s) |
> | -- | -- | -- | -- |
> | LiveStar  | 12.5   | 24.2 | 1.64 |
> | LiveCC_7B_Base  | 60.7  | 5.02 | 0.46  |
> | LiveCC_7B_Instruct  | 86.9  | 6.65 | 0.47  |
> | Proact-VL (LiveCC_Base, τ=0.3) | 54.2 | 5.13 | 0.40  |
> | Proact-VL (LiveCC_Base, τ=0.5) | 37.7 | 5.19 | 0.40  |
>
> To further understand the latency of proactive/event-triggered methods, we also analyze LiveStar under different `nums_run` values:
>
> | `nums_run` | Decision Time (s) | Generation Time (s) | Total Time (s) |
> | - | -| - | - |
> | 1    | 0.06 | 1.04   | 1.10  |
> | 5 (default) | 0.39   | 1.25   | 1.64 |
> | 10  | 0.92  | 1.18 | 2.10  |
>
> These results show that increasing `nums_run` mainly increases the decision overhead, which in turn increases total latency. This helps explain why heavier proactive/event-triggered methods are less practical for real-time interaction.

---

> > ### Author Rebuttal · Reviewer_Avhb · 2026-04-01
> >
> > Thanks for providing more details in dataset construction and add latency comparison. My concerns are addressed and decide to keep my original score "Weak Accept".

---

> > > ### Author Response · Authors · 2026-04-02
> > >
> > > We thank the reviewer for the encouraging and constructive feedback. We are glad that the additional details on dataset construction and latency comparison have addressed the concerns.

---

### Official Review · Reviewer_K8Zr · 2026-03-10

**Soundness:** 3
**Presentation:** 3
**Significance:** 3
**Originality:** 3
**Overall Recommendation:** 5
**Confidence:** 3

**Summary:**

This paper studies proactive real-time video assistants, instantiated as gaming commentators and player guides.
The authors argue that practical AI companions must solve three coupled problems at once: low-latency streaming perception, autonomous decisions about when to speak, and control over the length and pacing of each response.
To support this setting, the paper introduces the Live Gaming Dataset/Benchmark, built from 561 hours of English gameplay videos spanning 12 games and three scenarios: solo commentary, co-commentary, and user guidance.
The proposed method, Proact-VL, processes video in 1-second chunks together with optional query and history/context, uses a dedicated response head on a special FLAG token to decide whether to speak, and then generates short clip-level utterances when triggered.
Training combines standard language modeling with transition-aware and stability-oriented losses for speaking behavior, while inference uses a sliding-window KV-cache with reverse-RoPE style rebasing for long streams.
Across the proposed benchmark, a streaming benchmark, Ego4D style commentary, an unseen game setting, and a user study, the paper reports improved response timing and commentary quality over prior proactive and streaming baselines.

**Compliance With Llm Reviewing Policy:**

Affirmed.

**Key Questions For Authors:**

1. What validation protocol was used to select the response threshold and other important hyperparameters? Also, could the authors clarify whether the response head uses the final-layer or penultimate-layer FLAG hidden state? A clear response would improve reproducibility and reduce concern about hidden tuning or implementation ambiguity.

2. Can the authors provide more evidence that the LLM-judge metrics correlate well with human judgments for both timing and text quality? If the correlation is strong, that would significantly strengthen the confidence in the quantitative conclusions.

3. How is train/test separation enforced beyond video-wise splitting? In particular, are commentator identities, channels, or recurring source series separated across train and test? Stronger guarantees here would increase the confidence that the results reflect genuine generalization rather than source/style overlap.

**Limitations:**

yes

**Strengths And Weaknesses:**

Strengths:

1. [Significance] The paper tackles an important and timely problem. Real-time multimodal assistants need not only to understand video, but also to decide when to speak and how much to say. I find this problem formulation compelling, and the gaming setting is a practical and well-motivated testbed for studying proactive interaction.

2. [Originality] Although the paper does not introduce a radically new backbone architecture, the overall contribution is meaningfully original at the level of task formulation, benchmark construction, and system design. In particular, the paper combines chunk-wise streaming perception, explicit response triggering, and long-horizon inference into a coherent framework for short, proactive interaction.

3. [Soundness] The explicit response head for deciding whether to speak is well motivated and more controllable than relying on silence to emerge from ordinary decoding. The response-loss design is also sensible: emphasizing transition steps and regularizing speaking stability directly addresses the temporal structure of the task. The ablation on the training loss supports the usefulness of these design choices.

4. [Soundness] The empirical coverage is broad. Beyond the main benchmark, the paper includes long-horizon streaming evaluation, experiments on additional commentary settings, comparisons across different backbones, robustness checks for LLM-as-a-judge, and a user study. This makes the empirical section substantially stronger than a minimal benchmark-only evaluation.

5. [Presentation] The paper is generally well written and easy to follow. The motivation is clear, the figures are helpful, and I appreciate that the authors include concrete failure cases and a limitations section rather than presenting the system as uniformly successful.

Weaknesses:

1. [Soundness] A substantial part of the data construction and evaluation pipeline depends on model-generated supervision and LLM-based judging. While the paper includes robustness checks and a user study, additional human validation of the benchmark annotations and stronger evidence that the automatic metrics correlate with human judgments would further strengthen the empirical claims.

2. [Soundness/Presentation] It is not fully clear whether all proactive and real-time baselines were retrained or equivalently adapted under matched conditions. This does not invalidate the results, but it makes the magnitude of the empirical advantage somewhat harder to interpret.

3. [Presentation/Reproducibility] Some important details are deferred to the appendix, including parts of the metric definitions, hyperparameter choices, and implementation details that are central to reproducing the results. Consolidating a few of these details in the main paper would make the work easier to evaluate and reproduce.

4. [Significance/Claim Scope] The strongest evidence is currently in gaming-style commentary and guidance, so the broader claim about general AI companions would be strengthened by additional validation in non-gaming domains or by slightly narrowing the scope of the claim.

---

> ### Author Rebuttal · Authors · 2026-03-31
>
> We sincerely thank the reviewer for the careful and encouraging feedback. We address the main concerns below.
>
> > Question 1 & Weakness 3: Implementation Details and Reproducibility
>
> We thank the reviewer for pointing this out. In the revision, we will move key information currently deferred to the appendix—including metric definitions, hyperparameter choices, and implementation details—into the main paper.
>
> Regarding the response threshold, Appendix I.1 reports a threshold sweep and shows that a moderate range is broadly acceptable, with slightly different trade-offs between response frequency and response quality. In the main experiments (Secs. 5.1–5.4), we use τ = 0.3 as the default operating point, where the model’s overall response rate is around 50%.
>
> For several key hyperparameters, we provide analysis in Appendix I. Other hyperparameters are chosen based on training stability and empirical performance. For example, we set the response-loss weight to 0.2 so that the language modeling loss and the response loss stay at comparable scales during training.
>
> The response head uses the penultimate-layer <|FLAG|> hidden state. We apologize for the typo in the main text and will correct it in the revision. We use the penultimate layer following prior practice[2] that intermediate representations can sometimes provide stronger features than the final layer; this is also consistent with designs such as LLaVA-1.5[1]. We are also committed to open-sourcing the training, evaluation, and inference code.
>
> > Weakness 1 & Question 2: Validation of benchmark construction and LLM-based evaluation
>
> For benchmark construction, we do not rely on raw model outputs directly. For both ASR-derived commentary and tutorial-style QA, we apply a human-in-the-loop pipeline: manual spot-checking against the original video, LLM-based review for safety, consistency, and text quality, automatic flagging of suspicious samples, and final human review of flagged cases. We will make this protocol more explicit in the revision.
>
> We agree that stronger evidence on the alignment between automatic metrics and human judgments would strengthen the empirical claims. In the current paper, the user study and the main automatic results support the same overall conclusion. While they are not expected to match one-to-one, they provide complementary evidence with a consistent practical takeaway. We are currently conducting additional human studies and will update the rebuttal accordingly.
>
> > Question 3: Train/test separation and source overlap
>
> Our train/test split is video-level: each full video is assigned to a single split before clip segmentation, and in the multi-commentary setting, speaker separation is also performed only after this split assignment. This prevents direct content leakage across train and test, and the video content itself is strictly disjoint across splits. We acknowledge that the current split is not fully commentator-/channel-disjoint, since some single-player game videos may reflect one commentator’s style; however, Proact-VL still maintains a clear advantage on the unseen-game Wukong test set, where the game is used only for testing. We will clarify these split details in the revision.
>
> > Weakness 2: Clarification on baseline adaptation and fairness of comparison
>
> For all proactive and real-time baselines. To keep the comparison fair, we adapt the baselines under a standardized inference protocol and provide the same test-time information to all methods as much as their interfaces allow. We will clarify these adaptation details in the revision, so that the comparison setting is more transparent.
>
> > Weakness 4: Scope of the claim beyond gaming
>
> We agree that the current strongest evidence is in gaming-style commentary and guidance, and we will make this scope clearer in the revision. At the same time, Proact-VL does not simply lose general capability: Appendix H already reports results on a general benchmark, and we further evaluate on Video-MME (8 frames, without subtitles), where Proact-VL largely preserves the base model’s general video understanding capability and in some settings improves it:
>
> | Model                 | Short | Medium | Long | Overall |
> | --------------------- | ----- | ------ | ---- | ------- |
> | Qwen3VL               | 66.4  | 55.4   | 49.9 | 57.3    |
> | Proact-VL_{Qwen3VL}   | 67.6  | 58.2   | 51.4 | 59.1    |
> | Qwen2.5VL             | 61.9  | 51.2   | 46.2 | 53.1    |
> | Proact-VL_{Qwen2.5VL} | 64.8  | 56.8   | 49.7 | 57.1    |
>
> [1] Improved Baselines with Visual Instruction Tuning, CVPR 2024
> [2] Layer by Layer: Uncovering Hidden Representations in Language Models, ICML 2025

---

> > ### Author Rebuttal · Reviewer_K8Zr · 2026-04-02
> >
> > Thank you for the detailed and helpful clarifications. The rebuttal addresses my main concerns. I appreciate the authors’ commitment to clarifying these points in the revision. Stronger human validation of the automatic metrics and more explicit details on baseline adaptation in the final version would further strengthen the paper.

---

> > > ### Author Response · Authors · 2026-04-02
> > >
> > > We sincerely thank the reviewer for the positive feedback. We are glad that our rebuttal has addressed the main concerns.
> > > We also appreciate the suggestion on stronger human validation. We are currently organizing participant recruitment and will conduct human evaluation focusing on both response timing and text quality. The results will be used to assess agreement with the automatic judgments and will be reported in the final version.

---

### Official Review · Reviewer_oRCi · 2026-03-11

**Soundness:** 3
**Presentation:** 3
**Significance:** 3
**Originality:** 3
**Overall Recommendation:** 5
**Confidence:** 3

**Summary:**

- This paper introduces Proact-VL, a framework designed to transform multimodal LLMs into proactive interactive agents. The authors also present the Live Gaming Benchmark, which evaluates two gaming scenarios: game commentary and guidance. Experimental results demonstrate that Proact-VL achieves superior performance on the proposed benchmark in terms of both response quality and timing.

**Compliance With Llm Reviewing Policy:**

Affirmed.

**Final Justification:**

I think the main concern about the potential domain bias is addressed in the authors' rebuttal. Therefore, I will raise my score to 5.

**Key Questions For Authors:**

- What was the specific value of the threshold used for the experiments in Sections 5.1–5.5? How do you choose a suitable value?

**Limitations:**

yes

**Strengths And Weaknesses:**

Strengths
- The proposed proactive method is interesting, which introduing a specialized token to compute a "speaking probability".
- The model performance on Live Gaming Benchmark is impressive.

Weaknesses
- Potential domain bias and evaluation fairness. The evaluation is primarily conducted on the authors' self-constructed Live Gaming Benchmark. While the results are good, there is a concern regarding domain bias. Baseline models such as Live-CC or even GPT-4o may underperform simply because they are unfamiliar with the specific games in this new benchmark. Since Proact-VL is fine-tuned specifically on this data, it naturally possesses an internal advantage.  To provide a more objective evaluation of Proact-VL’s proactive dialogue capabilities, the authors should provide more convicing results. For instance, provide results on established open-source benchmarks, or conduct a generalization analysis using a test set of entirely unseen games.
- It is unclear whether the proposed specialized training for proactive responses degrades the model's video understanding performance. This paper does not report performance on general video understanding benchmarks like Video-MME.
- The font size in the figures (e.g., Figures 2 and 3) is too small, which is difficult to read.

---

> ### Author Rebuttal · Authors · 2026-03-31
>
> We thank the reviewer for the careful reading and constructive feedback. The comments are helpful for clarifying our evaluation and presentation, and we respond point by point below.
>
> > Weakness 1: Potential domain bias and evaluation fairness.
>
> We agree that evaluation fairness is important, especially since Proact-VL is fine-tuned with gaming-related data.
>
> First, our main results already include evaluation on Wukong set, which is used **only for testing and never for training**. The full results are reported in Appendix C.5, Table 10 (Full Results of Common and General Commentary). On this unseen game, Proact-VL still outperforms the baselines, although the margin is smaller than that on seen-game settings. This suggests that gaming-domain fine-tuning brings some domain-specific advantage, but the improvement cannot be explained by domain familiarity alone.
>
> Second, in Appendix F (Ablation Study for Training Data, we find that Response Quality improves across the training-data settings we tested, regardless of which data are used for training. This shows that the gain is **not tied to a particular game dataset** and that our method itself improves the model’s **proactive response capability**.
>
> Together, these results suggest that the gain comes from **both** domain-specific fine-tuning **and** the proposed proactive paradigm, rather than from game-specific memorization alone.
>
> > Weakness 2: Whether proactive training degrades general video understanding ability.
>
> We acknowledge that the current evaluation does not fully cover this aspect. We already report results on MVBench in Appendix H (Offline Baseline) as an initial evaluation of general video understanding ability, and following the reviewer’s suggestion, we further add experiments on broader general video understanding benchmarks. As shown in the two tables below, after proactive fine-tuning, Proact-VL shows small fluctuations on general video understanding datasets: it improves in some settings and is slightly lower in others, while overall largely preserving the base model’s general video understanding capability.
>
> **Results on Video-MME (8 frames) under different settings**
>
> | Model                 | Subs | Short | Medium | Long | Overall |
> | --------------------- | ---- | ----- | ------ | ---- | ------- |
> | Qwen3VL               | -    | 66.4  | 55.4   | 49.9 | 57.3    |
> | Proact-VL_{Qwen3VL}   | -    | 67.6  | 58.2   | 51.4 | 59.1    |
> | Qwen3VL               | ✓    | 68.0  | 56.7   | 50.3 | 58.3    |
> | Proact-VL_{Qwen3VL}   | ✓    | 68.9  | 57.4   | 53.1 | 59.8    |
> | Qwen2.5VL             | -    | 61.9  | 51.2   | 46.2 | 53.1    |
> | Proact-VL_{Qwen2.5VL} | -    | 64.8  | 56.8   | 49.7 | 57.1    |
> | Qwen2.5VL             | ✓    | 64.0  | 52.8   | 47.6 | 54.8    |
> | Proact-VL_{Qwen2.5VL} | ✓    | 67.1  | 57.4   | 50.8 | 58.4    |
>
> **Results on LongVideoBench (8 frames) under different settings**
>
> | Model               | 15   | 60   | 600  | 3600 | Overall |
> | ------------------- | ---- | ---- | ---- | ---- | ------- |
> | Qwen3VL             | 70.4 | 64.0 | 55.1 | 45.7 | 54.5    |
> | Proact-VL_{Qwen3VL} | 66.7 | 68.6 | 51.0 | 44.7 | 52.8    |
>
> Overall, these results indicate that proactive fine-tuning introduces only limited fluctuations on general benchmarks while preserving the base model’s general video understanding ability.
>
> > Weakness 3: Font size readability in the figures.
>
> Thank you for pointing this out. We will enlarge the font size in figures and further refine the layouts in the revision to improve readability.
>
> > Question 1: Threshold value used in Sections 5.1–5.5.
>
> For Sections 5.1–5.4, we use a response threshold of 0.3. For other experiments, we use 0.5. This is stated in Appendix C.3 (Experimental Details, Evaluation).
>
> We further analyze the effect of threshold choice in Appendix I.1 (Response Threshold). The results show a clear trade-off: a **lower threshold** makes the model more likely to speak, which can improve some text-quality-related judgments, but may also lead to overly frequent responses; a **higher threshold** makes the model more conservative, which can improve response appropriateness in some cases but reduce response frequency.
>
> Based on the sensitivity analysis in Appendix I.1, we use 0.3 for the main results. In particular, when τ = 0.3, the model’s overall response rate is around **50%**, which provides a relatively balanced operating point between responsiveness and response quality in our main interactive scenarios. For ablation and qualitative analyses, we use a fixed threshold of 0.5 so that these comparisons are conducted under the same setting.

---

> > ### Author Rebuttal · Reviewer_oRCi · 2026-04-02
> >
> > Good rebuttal. My concerns are all addressed.

---

> > > ### Author Response · Authors · 2026-04-02
> > >
> > > We sincerely thank the reviewer for the positive feedback and for updating the score. We are glad that our responses have addressed the concerns.

---

### Official Review · Reviewer_imv1 · 2026-03-16

**Soundness:** 3
**Presentation:** 3
**Significance:** 2
**Originality:** 1
**Overall Recommendation:** 4
**Confidence:** 4

**Summary:**

This paper introduces Proact-VL, a proactive VideoLLM framework for real-time streaming that addresses the challenge of determining when to speak and what to say by utilizing a novel Live Gaming Dataset and a chunk-wise architecture with an integrated trigger mechanism. By enabling the model to autonomously decide when to generate commentary, the proposed method outperforms existing baselines in both response timing and content quality.

**Compliance With Llm Reviewing Policy:**

Affirmed.

**Final Justification:**

I find the idea of incremental, cross-timestep speaking to be interesting and meaningful for real-time AI companions. Based on the rebuttal, I am inclined to raise my rating to Weak Accept.

However, I still consider the technical novelty to be somewhat limited. While the overall system is well-designed, many of the individual components appear to build on existing techniques, and the main contribution seems to be in the integration and engineering effort required to realize the proposed paradigm.

**Key Questions For Authors:**

Please explain novelty compared to related papers as written in the weakness section. I will raise the rating if the question is answered reasonably.

**Limitations:**

yes

**Strengths And Weaknesses:**

Strength
1. This paper addresses challenge in real-time multimodal interaction, determining the optimal timing for responses, offering a systematic solution for the emerging field of real-time AI companions.
2. By constructing the large-scale Live Gaming Dataset and Benchmark, the authors provide a valuable resource for research into proactive interaction and real-time video understanding.
3. The comprehensive experimental design effectively validates the framework against strong baselines across multiple dimensions, including text quality, response timing, and long-term streaming inference.


Weakness
1. Limited methodological novelty. While the paper presents a system that integrates chunk-wise streaming processing, a response trigger mechanism, and specialized training losses, the overall methodological contribution appears incremental. The core design, predicting whether the model should speak by applying a lightweight classification head on a special token and then generating text if triggered, is conceptually similar to existing event-triggered or proactive response frameworks in streaming video-language models. Many of the components (e.g., chunk-wise video processing, KV-cache streaming inference, and trigger-based response generation) have been explored in prior work on streaming VLMs and proactive dialogue systems. As a result, the primary contribution seems to lie more in system integration and engineering rather than introducing a fundamentally new modeling paradigm or algorithmic innovation.

2. Limited task generalization despite broad claims. Although the title and motivation emphasize “Real-Time AI Companions,” the actual task formulation and evaluation are heavily centered on gaming scenarios, particularly game commentary and gameplay guidance. The dataset, benchmarks, and experiments are all constructed around game streams and esports-style commentary, which represent a relatively narrow domain.

---

> ### Author Rebuttal · Authors · 2026-03-31
>
> We sincerely thank the reviewer for the detailed and constructive feedback. We clarify the concerns on novelty and task generalization below, and will revise the paper accordingly.
>
> > Weakness 1 & Key Question: Limited methodological novelty
>
> We agree that our work does not introduce a new VLM backbone; however, this represents a deliberate design choice. In the era of foundation models, we believe that principled methodological contributions are often more impactful than architectural complexity. By maintaining an architecture-agnostic approach, our method ensures superior flexibility and can be seamlessly integrated into various state-of-the-art backbones—a significant merit for scalability and practical adoption. Our contribution lies in a new formulation of real-time proactive interaction, together with the training and inference design needed to make it work in practice. We believe the novelty of our work lies in three aspects:
>
> ### (1) Novelty in the interaction paradigm
>
> Prior proactive or event-triggered systems typically decide whether to respond at each time step and, once triggered, generate a complete response immediately. In contrast, **Proact-VL performs incremental response generation**: it emits only a small amount of text at each step and continues unfinished utterances in the next step. Thus, the response is distributed over time rather than concentrated into a one-shot output.
>
> This design is better suited for real-time interaction. Instead of generating a complete response at once, the model can **watch while speaking** and refine its response as new visual evidence arrives. By contrast, in many prior methods, response decision and generation are handled separately, making coherent continuation across consecutive timesteps more difficult. For example, methods such as LiveStar or Dispider may generate relatively long responses at a single trigger point, while MMDuet may produce repetitive content across adjacent timesteps.
>
> ### (2) Novelty in the training design tailored to the proposed paradigm
>
> This formulation creates new training challenges beyond conventional trigger-once-then-generate settings: the model must learn both **when to speak** and **how to continue speaking coherently across consecutive timesteps**. To support this, we design a dedicated training objective tailored to the proposed paradigm.
>
> ### (3) Novelty in the unified end-to-end system design
>
> Prior works often focus on different aspects of the problem, such as streaming perception, response timing, or response generation, but do not provide a unified and complete framework for real-time proactive interaction. In contrast, Proact-VL provides a unified end-to-end framework that connects data construction, interaction formulation, training, and practical streaming inference.
>
> > Weakness 2: Limited task generalization despite broad claims
>
> We agree that the current evaluation is mainly centered on gaming scenarios, and we will revise the paper to make this scope clearer. In this work, we use gaming as a concrete and demanding testbed for studying proactive response timing and real-time multimodal interaction.
>
> To avoid over-specializing to gaming only, our training set also includes Ego4D Goal-Step and part of Live-WhisperX-526K, which provide more general egocentric and commentary-style video data. We also evaluate Proact-VL on Ego4D (Table 3) and MVBench (Appendix H).
>
> To further address this concern, we additionally evaluate Proact-VL against the base model on Video-MME and LongVideoBench. For Video-MME, we uniformly sample 8 frames from each video and evaluate with and without subtitles. The results show that Proact-VL largely preserves the base model’s general-domain video understanding ability, with slight improvements on Video-MME. On LongVideoBench, the fine-tuned model shows a small drop, but the gap remains limited and acceptable.
>
> #### Additional results on Video-MME (8-frame)
>
> | Model              | Subs | Short | Medium | Long | Overall |
> | -- | ---- | ----- | ---- | --- | --- |
> | Qwen3VL       | -    | 66.4  | 55.4   | 49.9 | 57.3    |
> | Proact-VL_{Qwen3VL}   | -    | 67.6  | 58.2   | 51.4 | 59.1 |
> | Qwen3VL       | ✓    | 68.0  | 56.7   | 50.3 | 58.3    |
> | Proact-VL_{Qwen3VL}   | ✓    | 68.9  | 57.4   | 53.1 | 59.8 |
> | Qwen2.5VL    | -    | 61.9  | 51.2   | 46.2 | 53.1    |
> | Proact-VL_{Qwen2.5VL} | -    | 64.8  | 56.8   | 49.7 | 57.1 |
> | Qwen2.5VL    | ✓    | 64.0  | 52.8   | 47.6 | 54.8    |
> | Proact-VL_{Qwen2.5VL} | ✓    | 67.1  | 57.4   | 50.8 | 58.4 |
>
> #### Additional results on LongVideoBench (8-frame)
>
> | Model  | 15   | 60   | 600  | 3600 | Overall |
> | -- | -- | -- | -- | -- | -- |
> | Qwen3VL             | 70.4 | 64.0 | 55.1 | 45.7 | 54.5    |
> | Proact-VL_{Qwen3VL} | 66.7 | 68.6 | 51.0 | 44.7 | 52.8    |
>
> We will include these additional results and implementation details in the revision, and clarify that gaming is the primary testbed in this work rather than the only intended application scenario.

---

> > ### Author Rebuttal · Reviewer_imv1 · 2026-04-04
> >
> > Thank you for the detailed response. I appreciate the clarification on the proposed interaction paradigm and the additional experiments provided.
> >
> > I find the idea of incremental, cross-timestep speaking to be interesting and meaningful for real-time AI companions. Based on the rebuttal, I am inclined to raise my rating to Weak Accept.
> >
> > However, I still consider the technical novelty to be somewhat limited. While the overall system is well-designed, many of the individual components appear to build on existing techniques, and the main contribution seems to be in the integration and engineering effort required to realize the proposed paradigm.

---

> > > ### Author Response · Authors · 2026-04-07
> > >
> > > We thank the reviewer for the thoughtful and helpful feedback. We are encouraged that the reviewer finds the idea interesting and meaningful for real-time AI companions, and we appreciate the positive update in the rating.
> > >
> > > We understand the reviewer’s perspective regarding the level of technical novelty. While our method builds on some existing techniques, we see the contributions of this work mainly in the formulation of this paradigm and in making it trainable and effective in practice.
> > >
> > > More broadly, we view this work as an exploration of streaming video understanding in real-time interactive settings, which remains a challenging and relatively underexplored problem. As the reviewer pointed out, developing more substantially novel and more effective methods for this setting remains an important direction, and this is also a key direction of our future work. We hope this work can provide useful insights for future research in this area.

---

### Decision · Program_Chairs · 2026-04-30

**Decision:**

Accept (regular)

**Comment:**

This paper introduces Proact-VL, a framework for real-time, proactive multimodal agents, alongside the Live Gaming Benchmark, and has received consistently positive feedback from all reviewers (with final scores of two Accepts and two Weak Accepts). The reviewers commended the paper for tackling a highly practical and timely problem, highlighting the novel "chunk-wise" interaction paradigm that distributes responses over time, the strong system engineering via an explicit response head for pacing control, and the utility of the proposed benchmark. During the review phase, initial concerns were raised regarding incremental methodological novelty, potential domain bias from the gaming-centric training data, reliance on LLMs for data annotation, and the absence of direct latency comparisons. In a comprehensive rebuttal, the authors successfully mitigated these issues by demonstrating that the model retains its general video understanding capabilities on broad benchmarks like Video-MME, detailing their rigorous human-in-the-loop data pipeline, and supplying new latency metrics that prove Proact-VL's real-time viability compared to baselines. While one reviewer maintained that the foundational architectural novelty is somewhat limited, they agreed that the system integration is highly effective and meaningful raising their final score. Given the solid technical execution, the practical value of the benchmark, and the authors' effective resolution of the primary critiques, this paper is recommended for acceptance.